# Beyond Single Concept Vector: Modeling Concept Subspace in LLMs with Gaussian Distribution

**Haiyan Zhao**[1]**, Heng Zhao**[2]**, Bo Shen**[1]**, Ali Payani**[3]**, Fan Yang**[2]**, Mengnan Du**[1,*]
[1]New Jersey Institute of Technology, [2]Wake Forest University, [3]Cisco Research
{hz54, bo.shen, mengnan.du}@njit.edu, {zhaoh21, yangfan}@wfu.edu, apayani@cisco.com
[*]Corresponding author

## Abstract

Probing learned concepts in large language models (LLMs) is crucial for understanding how semantic knowledge is encoded internally. Training linear classifiers on probing tasks is a principle approach to denote the vector of a certain concept in the representation space. However, the single vector identified for a concept varies with both data and training, making it less robust and weakening its effectiveness in real-world applications. To address this challenge, we propose an approach to approximate the subspace representing a specific concept. Built on linear probing classifiers, we extend the concept vectors into Gaussian Concept Subspace (GCS). We demonstrate GCS's effectiveness through measuring its faithfulness and plausibility across multiple LLMs with different sizes and architectures. Additionally, we use representation intervention tasks to showcase its efficacy in real-world applications such as emotion steering. Experimental results indicate that GCS concept vectors have the potential to balance steering performance and maintaining the fluency in natural language generation tasks.

## 1 Introduction

Large language models (LLMs) such as GPT-4 (Achiam et al., 2023), LLaMA-3 (Touvron et al., 2023), and Claude-3 (Anthropic, March 4, 2024) have demonstrated remarkable capabilities across a wide range of natural language understanding and generation tasks (Brown et al., 2020; Wei et al., 2022). However, our understanding of how concepts are represented internally within these models remains limited. Current research in this area can be broadly categorized into two categories. The first involves studying model parameters through causal mediation analysis (Vig et al., 2020), which has found applications in downstream tasks such as knowledge editing (Meng et al., 2022; Wang et al., 2024) and circuit identification (Conmy et al., 2023; Elhage et al., 2021). The second focuses on investigating the representation space and activations of models (Zou et al., 2023; Turner et al., 2023; Jorgensen et al., 2023; Rimsky et al., 2023), where representation vectors in hidden spaces are often interpreted as linear vectors of certain concepts, such as color (Patel & Pavlick, 2022), world models (Nanda et al., 2023; Li et al., 2023), and sentiment (Tigges et al., 2023).

In this work, we focus on the representation vector-based explanation paradigm. Among methods to derive representation vectors, linear concept vectors are most widely used. They are typically obtained by training linear probing classifiers, such as logistic regression, on probing datasets comprising positive and negative samples (Ousidhoum et al., 2021; Alain & Bengio, 2016). This approach has proven effective in identifying semantic knowledge encoded within LLMs and has recently been extensively applied in inference-time interventions to mitigate undesirable model behaviors (Li et al., 2024; Lee et al., 2024; Rimsky et al., 2023; Turner et al., 2023). However, concept vectors derived from this approach can vary significantly depending on the probing dataset used to train the classifier and the training process, making it challenging to obtain robust vectors. This substantial variation not only poses challenges in studying concept relations within the representation space but also negatively impacts effectiveness of concept vector's usage in intervention tasks.

To address these limitations, we explore the following research question: *Instead of using a single vector to describe each underlying concept captured by LLMs, can we use a distribution to more robustly describe that learned concept?* This is motivated by the observation that the representation of a concept may be multifaceted, potentially constituting a multidimensional subspace in the representation space. Building upon this assumption, our work focuses on approximating a subspace that better captures the semantics of specific concepts while providing vectors representing a certain concept with varying relevance.

Based on this motivation, we propose Gaussian Concept Subspace (GCS), which improves upon linear probing classifiers described by Ousidhoum et al. (2021); Kim et al. (2018); Alain & Bengio (2016) by using a Gaussian distribution to estimate the potential directional subspace of a concept. The main idea is to learn a set of vectors termed *observed vectors* instead of an individual vector for each concept. These observed vectors are then used to estimate the Gaussian distribution of the concept subspace. The vectors within this Gaussian distribution represent varying degrees of relevance to the concept, with vectors closer to the mean considered more closely related, termed *sampled vectors*. This method offers a more nuanced representation of concepts, capturing their multifaceted nature within a multidimensional subspace rather than reducing them to a single vector.

We conduct experiments to evaluate the effectiveness of GCS from two perspectives: 1) Whether concept vectors sampled from GCS effectively describe a given concept learned by LLMs; and 2) Whether concept vectors derived from GCS are useful in downstream interventions to guide LLMs to generate more desirable outputs. For the first perspective, we evaluate sampled vectors using two categories of metrics: faithfulness and plausibility. For faithfulness, we examine similarities within observed vectors, sampled vectors, and between both sets to demonstrate their relatedness in the representation space. For plausibility, we investigate whether the explanations align with our understanding of inter-concept relations. For the second perspective, we compare the performance of sampled vectors on inference-time intervention tasks with that of vectors derived with mean difference and single training classifier. Specifically, we applied these vectors to steer models towards generating more joyful movie reviews. Our results demonstrate that sampled vectors from GCS are both faithful and plausible in terms of their explanations. Moreover, they outperform vectors from other methods in manipulating models to generate desirable content while maintaining text fluency.

## 2 PRELIMINARY

**Hidden Representation.** We focus on decoder-only LLMs (GPT-like models) (Ferrando et al., 2024), where each layer comprises multi-head attention blocks (MHA) and feed-forward networks (FFNs/MLPs). In this study, we utilize frozen pretrained language models. We index layers as $\ell \in L$, where $L$ denotes the set of all model layers. Each layer begins and ends with a residual stream. The MHA first processes the residual stream and adds its output back to it. The updated vector is then passed through MLPs to generate the layer's output:

$$\boldsymbol{h}_i^{\ell+1} = \boldsymbol{h}_i^\ell + \text{Att}^\ell\left(\boldsymbol{h}_i^\ell\right) + \text{MLP}^\ell\left(\boldsymbol{h}_i^\ell + \text{Att}^\ell\left(\boldsymbol{h}_i^\ell\right)\right), \tag{1}$$

where $\boldsymbol{h}_i^\ell$ represents the $i$-th token in the input token sequence at layer $\ell$. We concentrate on the output representation space of each layer, specifically the residual stream at the layer's end. Following Zou et al. (2023), we use the last token's representation to represent the entire input sequence, denoted as $\boldsymbol{h}^\ell$, as it is generally believed to integrate information from all preceding tokens.

**Linear Probing.** We adopt the linear concept probing approach to derive concept activation vectors (Kim et al., 2018). Following prior work (Li et al., 2024), for each concept $c$, given a probing dataset $\mathcal{D}_c = \{(\boldsymbol{h}_i, y_i) \mid i = 1, \cdots, N\}$ where $\boldsymbol{h} \in \mathbb{P} \cup \mathbb{N}$, $\mathbb{P}$ is the set of concept-related positive samples, $\mathbb{N}$ is the set of concept-irrelevant negative samples, and $y_i \in \{0, 1\}$ indicates the sample class. These datasets generate layerwise hidden representations for downstream classifier training. We employ a logistic regression (LR) classifier as: $\boldsymbol{\sigma}(\boldsymbol{h}_c^\ell) = \frac{1}{1+\exp(-\boldsymbol{h}_c^\ell \cdot \boldsymbol{w}_c^\ell)}$, where $\boldsymbol{h}_c^\ell \in \mathbb{R}^{N \times d}$ specifies sample representations from a probing dataset at $\ell$-th layer, and $d$ is the representation dimension. The normalized coefficients $\boldsymbol{w}_c^\ell$ of each binary linear classifier are considered the concept vector. The loss function for logistic regression with L2 regularization is expressed as follows:

$$\hat{\boldsymbol{w}}_c^\ell = \underset{\boldsymbol{w}_c^\ell}{\text{argmin}} -\frac{1}{n}\sum_{i=1}^n y_i \log \boldsymbol{\sigma}(\boldsymbol{h}_i^\ell) + (1 - y_i) \log\left(1 - \boldsymbol{\sigma}(\boldsymbol{h}_i^\ell)\right) + \frac{\lambda}{2}|\boldsymbol{w}_c^\ell|_2^2, \tag{2}$$

where hyperparameter $\lambda$ controls the strength of regularization. However, the derived concept vector $\hat{\boldsymbol{w}}_c^\ell$ can vary substantially depending on the probing dataset $\mathcal{D}_c$ used to train the classifier $\boldsymbol{\sigma}(\boldsymbol{h}_c^\ell)$ and the specific training process. This variability makes it challenging to obtain robust and consistent vectors. Such instability not only complicates the study of concept relations within the representation space but also negatively impacts the effectiveness of intervention tasks that rely on these vectors.

## 3 METHODOLOGY

To address the above-mentioned limitations, we investigate whether representing concepts learned by LLMs as distributions, rather than single vectors. We aim to approximate a subspace that better captures the semantics of specific concepts while providing vectors representing the concept with varying relevance. Building upon the linear probing approach described in Section 2, we introduce the Gaussian Concept Subspace (GCS) framework for estimating a multidimensional subspace of vectors for each concept. While linear probing identifies a single vector for a concept, GCS aims to capture a more comprehensive representation by estimating a Gaussian distribution of vectors.

### 3.1 THE PROPOSED GCS FRAMEWORK

To estimate the Gaussian distribution of concept vectors, we extend the linear probing method by deriving $M$ independent concept vectors for a concept $c$. We first create a large dataset for a concept $c$ and then randomly sample subsets of examples from this dataset to create $M$ smaller probing datasets. Each probing dataset contributes to the same concept vector $c$, analogous to the single vector derived in Equation 2. For a model with target layer $\ell$, we obtain a concept vector $\boldsymbol{w}_{c,\, m}^\ell$ for the $m$-th probing dataset at the $\ell$-th layer. Please refer to Section 4.1.3 for more details of GCS framework. This process generates a set of vectors for concept $c$:

$$\boldsymbol{W}_c^\ell = \{\boldsymbol{w}_{c,\, m}^\ell \mid m = 1, \cdots, M\}. \tag{3}$$

We use this set to estimate the concept subspace, represented as a Gaussian distribution of concept $c$ at the $\ell$-th layer. The distribution is $d$-dimensional, where $d$ is the dimension of the hidden representations, and is denoted as:

$$\boldsymbol{W}_c^\ell \sim \mathcal{N}_d(\boldsymbol{\mu}_c^\ell, \boldsymbol{\Sigma}_c^\ell). \tag{4}$$

To simplify the model, we assume independence between dimensions in the representation space. This assumption results in a diagonal covariance matrix, where diagonal elements represent the variance of each dimension:

$$\boldsymbol{\mu}_c^\ell = \frac{1}{M} \sum \boldsymbol{w}_{c,\, m}^\ell; \quad \boldsymbol{\Sigma}_c^\ell = \mathrm{diag}(\sigma_1^2, \cdots, \sigma_d^2) \tag{5}$$

The mean vector $\boldsymbol{\mu}_c^\ell$ is calculated by averaging the corresponding dimensions of the $M$ observed concept vectors, which are essentially the weight vectors from the trained binary classifiers in the linear probing approach. This multivariate distribution describes a subspace around the observed concept vectors, extending the single-vector representation of linear probing to a multidimensional subspace. Concept vectors within this distribution can be sampled at various standard deviations (e.g., $1\sigma$, $2\sigma$, ..., $n\sigma$) from the mean vector. For our evaluations, we randomly sample vectors $\boldsymbol{V}_c^\ell$ (the set of multiple sampled vectors $\boldsymbol{v}_c^\ell$) within $1\sigma$ of the mean to represent the concept.

The overall steps of the GCS framework are summarized in Algorithm 1. By moving from a single vector to the Gaussian distribution of vectors, GCS aims to capture a more robust representation of concepts in the language model's hidden space, potentially offering improvements over the linear probing approach in both interpretability and downstream applications.

---

**Algorithm 1:** Proposed GCS Framework

**Input:** LLM to be explained, concept $c$, layer $\ell$, probing dataset number $M$.

**while** *first stage* **do**
    Generate $M$ probing datasets for concept $c$;
    **for** $m = 1$ **to** $M$ **do**
        Train classifier on dataset $m$ for $c$;
        Extract weight vector $\boldsymbol{w}_{c,m}^\ell$;
    **end**
    $\boldsymbol{W}_c^\ell = \{\boldsymbol{w}_{c,m}^\ell \mid m = 1, \ldots, M\}$;
**end**

**while** *second stage* **do**
    $\boldsymbol{\mu}_c^\ell = \frac{1}{M} \sum \boldsymbol{w}_{c,m}^\ell$;
    $\boldsymbol{\Sigma}_c^\ell = \mathrm{diag}(\sigma_1^2, \ldots, \sigma_d^2)$;
**end**

**while** *third stage* **do**
    Sample vector $\boldsymbol{v}_c^\ell \sim \mathcal{N}_d(\boldsymbol{\mu}_c^\ell, \boldsymbol{\Sigma}_c^\ell)$ in $1\sigma$;
**end**

**Output:** Sampled concept vectors $\boldsymbol{v}_c^\ell$.

---

### 3.2 EVALUATION OF GCS

We use faithfulness and plausibility two metrics to eveluate performance of GCS, which are two crucial dimensions for evaluating the performance of an interpretability approach (Zhao et al., 2024).

#### 3.2.1 FAITHFULNESS

To assess faithfulness, we compare the performance of sampled vectors $V_c^\ell$ (stage 3 of Algorithm 1) compared to that of observed vectors $W_c^\ell$ (stage 1 of Algorithm 1). Specifically, we employ two metrics: 1) similarity among observed vectors, sampled vectors and between the two sets; 2) accuracy of classifiers based on observed vectors and sampled vectors on our constructed datasets.

**Similarity**   We evaluate the similarity between vectors within the observed concept vectors $W_c^\ell$ and within the sampled concept vectors $V_c^\ell$ using Equation 6. Additionally, we assess the similarity between vectors from the observed set and the sampled set using Equation 7.

$$S_{\text{observed}}^\ell = 1/N * \sum_{i,j} \text{sim}\left(\boldsymbol{w}_i, \boldsymbol{w}_j\right), \ \boldsymbol{w}_i, \boldsymbol{w}_j \in W_c^\ell; \ S_{\text{sampled}}^\ell = 1/N * \sum_{i,j} \text{sim}\left(\boldsymbol{v}_i, \boldsymbol{v}_j\right) \boldsymbol{v}_i, \boldsymbol{v}_j \in V_c^\ell \quad (6)$$

$$S_{\text{O-S}}^\ell = 1/N * \sum_{i,j} \text{sim}\left(\boldsymbol{w}_i, \boldsymbol{v}_j\right), \ \boldsymbol{w}_i \in W_c^\ell, \ \boldsymbol{v}_j \in V_c^\ell \quad (7)$$

 The average similarity score of observed vectors $S_{\text{observed}}^\ell$ and sampled vectors $S_{\text{sampled}}^\ell$ describes how similar the linear vectors are for a specific concept. If our sampled vectors accurately represent concept vectors, the similarity score of sampled vectors should be comparable to that of observed concept vectors. Moreover, the similarity between observed and sampled vectors, denoted as $S_{\text{O-S}}^\ell$, should also be approximately equal to the within-set similarities.

**Accuracy**   The observed concept vectors are derived from the weights of linear concept classifiers. To generate $M$ linear concept vectors for a concept $c$, we randomly sample $M$ probing datasets $\{\mathcal{D}_1, \cdots, \mathcal{D}_M\}$ from a large dataset $\mathcal{D}$. For each probing dataset $\mathcal{D}_i$, a linear classifier is trained using hidden representations from the $\ell$-th layer of a LLM. The weight vector of this classifier is identified as a concept vector for concept $c$ at layer $\ell$. The average accuracy score of concept vectors for concept $c$ at the $\ell$-th layer is defined as:

$$A_O^\ell = \frac{\sum_{i=1}^M \text{acc}\left(\boldsymbol{w}_i, \mathcal{D}_{test}\right)}{M}, \text{ where } \mathcal{D}_{test} = \mathcal{D}/\mathcal{D}_i; \ A_S^\ell = \frac{\sum_{j=1}^M \text{acc}\left(\boldsymbol{v}_j, \mathcal{D}_{test}\right)}{M}, \text{ where } \mathcal{D}_{test} = \mathcal{D} \quad (8)$$

 The average accuracy score captures how effectively concept vectors represent the concept. $A_O^\ell$ and $A_S^\ell$ denote average accuracy score for observed vectors and sampled vectors respectively. A higher average accuracy score for observed concept vectors suggests that classifiers have learned more generalizable concept vectors. Sampled concept vectors should achieve comparable or even higher accuracy scores if they genuinely represent related concept.

#### 3.2.2 PLAUSIBILITY

Plausibility describes how well explanations align with human expectations. We measure the plausibility of GCS by studying whether models learn real-world hierarchies of concepts. To achieve this, we predefine a set of hierarchical concepts. We use average cosine similarity among concept vectors and principle component analysis (PCA) to demonstrate the consistence of GCS's plausibility.

**Average Cosine Similarity**   To measure the similarity between two concepts $c_i$ and $c_j$ at the $\ell$-th layer, we utilized their respective sampled concept vectors $V_{c_i}^\ell$ and $V_{c_j}^\ell$. The concept similarity is represented by the average cosine similarity between two concept vector sets. The average cosine similarity between these two concepts at the $\ell$-th layer is computed as follows:

$$S_{c_i, c_j}^\ell = \frac{\sum_{m=1}^M \sum_{n=1}^M \text{sim}\left(\boldsymbol{v}_m, \boldsymbol{v}_n\right)}{M \times M/2}, \boldsymbol{v}_m \in V_{c_i}^\ell, \boldsymbol{v}_n \in V_{c_j}^\ell. \quad (9)$$

**PCA visualization**   PCA aims to linearly reduce high-dimensional data onto lower dimensions. It excels in providing an intuitive understanding of relations between data points (Marks & Tegmark, 2023). We adopt PCA to visualize the relationship among predefined concepts. Since each concept has thousands vectors, we use the mean vector of sampled concept vectors to represent the concept.

## 4 EXPERIMENTS

In this section, we present experimental results that measure the performance of GCS and address the following three research questions (**RQs**):

**RQ1:** How faithfully do GCS-sampled concept vectors represent the original concepts?

**RQ2:** To what extent do explanations derived from GCS-sampled concept vectors align with human expectations about the hierarchies among model's learned knowledge?

**RQ3:** Can the proposed GCS method effectively mitigate unwanted behaviors in LLMs?

### 4.1 EXPERIMENTAL SETUP

#### 4.1.1 DATASETS

We define a hierarchical concept structure comprising two levels, encompassing concepts from high-level to low-level. The high-level concepts include four categories: `movie`, `sports event`, `populated place`, and `animal`. Under each high-level concept category, there are 4 low-level concepts, as illustrated in Figure 1. To ensure consistency and high quality of input data across all experiments, we generate our experimental datasets using the OpenAI API, specifically leveraging GPT-4o model. For each low-level concept, we prompt GPT-4o to produce 5,000 positive samples and 5,000 negative samples. The specific prompts used for generating these samples are detailed in Appendix J. Our prompt design strategy varies based on the concept's specificity. For more inclusive concepts such as "`Town`", we utilize detailed prompt descriptions to guide GPT-4o in generating concept-related data. Conversely, for more specific concepts like "`Motor race`", simpler and more generic prompt descriptions are sufficient to produce appropriate samples.

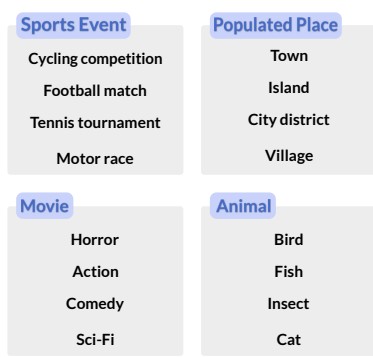

Figure 1: Hierarchical concepts.

#### 4.1.2 MODELS

We study concept vectors across multiple LLMs, including Llama-2-7B chat model (Meta AI, 2023b), Gemma-7B (Google, 2024), and Llama-2-13B chat model (Meta AI, 2023a). This selection of models enables a comparative analysis of GCS across varying model architectures and sizes. By examining GCS's performance across these diverse models, we aim to provide insights into the method's generalizability and effectiveness across different LLM implementations.

#### 4.1.3 IMPLEMENTATION DETAILS

Evaluating GCS at the distribution level presents significant challenges. Gaussian distribution encompasses vectors representing concepts with varying degrees of relevance, potentially diminishing as the sampling distance from the mean increases. This variability, coupled with the infinite potential of sampled concept vectors, renders precise distribution-level evaluation impractical. To address these limitations, we simplify our assessment by randomly sampling 1,000 concept vectors within $1\sigma$ of the mean, which we consider representative of the concept.

The GCS pipeline comprises several key steps: generating data points using GPT-4o, extracting hidden representations of each sample from LLMs, training linear classifiers for each concept using these hidden representations, and constructing Gaussians distribution for concept vectors. Our study generated 10,000 descriptions per low-level concept (5,000 positive and 5,000 negative). To efficiently utilize this data, we repeatedly sample 1,000 positive and 1,000 negative descriptions with replacement to create smaller, concept-specific probing datasets, each yielding an observed concept vector. This process is repeated 1,000 times for each low-level concept, resulting in 1,000 observed vectors per concept.

These observed vectors are used to derive the mean and covariance of the Gaussian distribution for each concept. To simplify computation, we assume independence between vector dimensions, re-

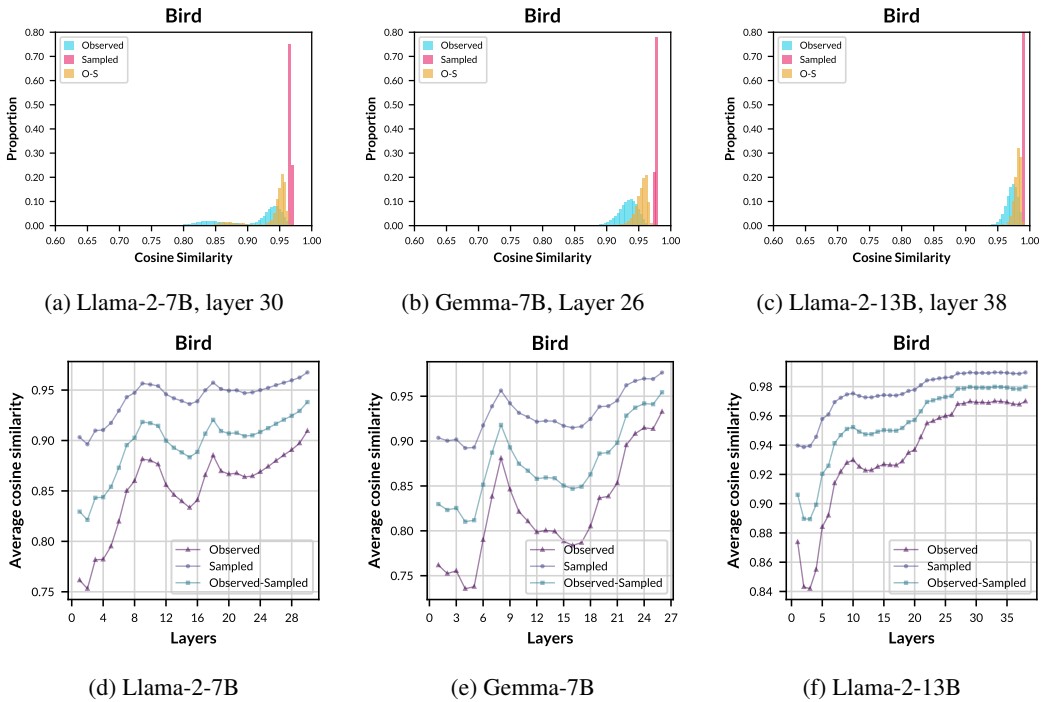

Figure 2: Histogram of cosine similarity within observed concept vectors, sampled concept vectors, and between both sets for concept "Bird" (a-c). Layer-wise average cosine similarity from the second layer to the penultimate layer of observed concept vectors, sampled concept vectors, and between both sets for concept "Bird" (d-f).

ducing the covariance matrix to a diagonal matrix where each diagonal value represents the variance of the concept vector in that dimension.

Notably, our classifiers are trained on hidden representations of each description. Following Zou et al. (2023), we use the hidden representation of the last token within a description from a specific layer as the representation for that description at that layer. We use L2 regularization when training the logistic regression classifier and the regularization weight $\lambda$ in Equation 2 is set as 1.0. Our evaluation primarily focuses on hidden representations from the penultimate layer of each model, as we observed improved concept learning in deeper layers (see Appendix E, F, and G).

## 4.2 FAITHFULNESS OF GCS (**RQ1**)

We evaluate the faithfulness of GCS from two perspectives: *similarity* and *accuracy*. Similarity assessment involves comparing observed concept vectors with those sampled from the GCS. This comparison provides an effective measure of their relatedness in the representation space. Ideally, for any given concept, the sampled vectors should closely approximate the observed vectors. We also assess the accuracy of these vectors in concept-related tasks.

### 4.2.1 COSINE SIMILARITY AMONG CONCEPT VECTORS

For each concept, we analyze three types of cosine similarity: among observed concept vectors, among sampled concept vectors, and between these two sets, as defined in Equation 6 and 7. The cosine similarities among observed concept vectors indicate the potential size of the concept subspace, with higher similarities suggesting a more compact subspace. The cosine similarities among sampled concept vectors reflect their diversity. Vectors sampled within $1\sigma$ are generally less diverse but closer to the center of the conceptual space. In this section, we focus on sampling concept vectors within the $1\sigma$ range, which are central to the concept subspace. The similarity results are shown in Figure 2, the key findings are summarized as follows.

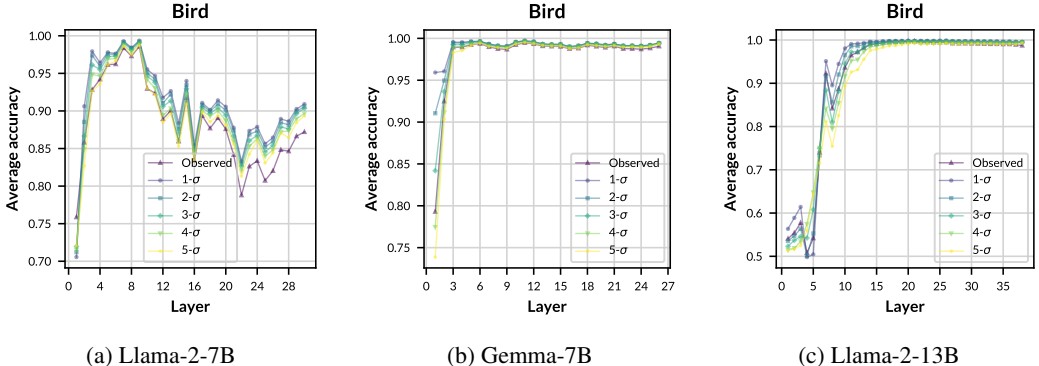

Figure 3: Accuracy of observed and sampled concept vectors aross varying models.

First, we investigate these three types of similarities across LLMs of varying sizes and structures, as shown in Figure 2 (a, b, and c) and Appendix E. Our results indicate that similarities among observed concept vectors exhibit greater variation than those among sampled concept vectors, ranging from approximately 0.8 to 0.9. Sampled concept vectors within the $1\sigma$ range consistently demonstrate very high cosine similarities among themselves, with cosine similarity larger than 0.93. This indicates that GCS produces more consistently related vectors than the original observations. Cosine similarities between observed and sampled concept vectors typically range from 0.88 to 0.93. Besides, the patterns of cosine similarity are remarkably consistent across different model sizes and architectures (Llama-2-7B, Gemma-7B, and Llama-2-13B). This suggests that the GCS method is robust across different language models.

Second, to better observe changes in concept vectors within each concept, we also examine the average cosine similarity of these concept vectors at the layer level as shown in Figure 2 (d, e and f) and Appendix F. There is a general upward trend in cosine similarity as we move through the layers. This indicates that concept representations become more refined and consistent in deeper layers of the models. The sampled vectors (blue lines in d, e, f) show the highest and most stable cosine similarity across layers. This indicates that the sampling vectors from GCS produce consistently related vectors throughout the model.

### 4.2.2 PREDICTION ACCURACY OF CONCEPT VECTORS

Despite the similarity between sampled and observed concept vectors in the representation space, it is crucial to verify that sampled vectors are as effective as observed concept vectors. We begin by measuring the accuracy of each observed linear vector on its test dataset, which comprises 8,000 additional concept descriptions. Then, we evaluate sampled concept vectors on the entire dataset of 10,000 samples, as these vectors are not trained on the dataset. The results are reported in Figure 3.

To evaluate the impact of proximity to the concept subspace center, we evaluate sampled concept vectors for all concepts at $1\sigma$, $2\sigma$, $3\sigma$, $4\sigma$, and $5\sigma$ across three LLM models. Our results demonstrate that sampled concept vectors usually achieve comparable and even better prediction accuracy compared to observed vectors, as illustrated in Figure 3. This improved performance is particularly evident for sampled concept vectors at $1\sigma$ and $2\sigma$. These findings are further corroborated by the accuracy of these vectors across all predefined concepts in the Gemma-7B model (Appendix G). Our analysis yields two key insights. First, it validates the effectiveness of sampled concept vectors in representing concepts. Second, it also suggests that vectors closer to the subspace center could be more representative of the concept.

### 4.3 PLAUSIBILITY OF GCS (**RQ2**)

In this section, we evaluate the plausibility of GCS using two methods: average cosine similarity among concepts and PCA visualization (see Section 3.2.2). Here, we utilize concept vectors sampled at $1\sigma$ to compute the average cosine similarity between concepts and to generate the PCA visualization. We assess the coherence of concept relationships in the high-dimensional space (through cosine similarities) and in a lower-dimensional projection (through PCA visualization).

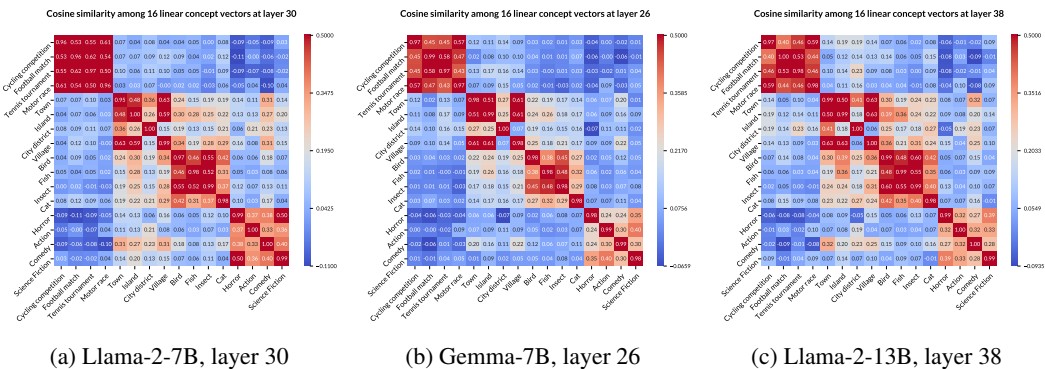

(a) Llama-2-7B, layer 30     (b) Gemma-7B, layer 26     (c) Llama-2-13B, layer 38

Figure 4: Heatmap of concept average cosine similarity of 16 concepts across Llama-2-7B, Gemma-7B, and Llama-2-13B. The 16 low-level concepts are grouped into four high-level categories: the first 4 rows/columns represent *sports events*, the next 4 represent *populated places*, followed by 4 for *animals*, and the last 4 for *movie genres*.

### 4.3.1 AVERAGE COSINE SIMILARITY AMONG CONCEPTS

We evaluate concept similarity using average cosine similarity. Instead of measuring the entire distribution, we assess 1,000 concept vectors sampled within $1\sigma$ for each concept. The cosine similarity is computed for all vector pairs between two concepts, with the mean value representing the overall similarity between these concepts. Figure 4 illustrates these results across various LLMs.

The analysis of concept cosine similarities across three different language models reveals several key insights. First, low-level concepts within the same high-level category typically exhibit higher similarity scores. All three models (Llama-2-7B, Gemma-7B, and Llama-2-13B) show similar overall patterns, with distinct $4 \times 4$ blocks along the diagonal representing high intra-category similarities. These intra-category similarities are generally higher compared to similarities with low-level concepts from other high-level categories. Second, while all high-level categories show internal coherence, some categories appear more tightly related than others. For example, the "sports events" category (top-left $4 \times 4$ block) consistently shows very high internal similarity across all models, suggesting these concepts are closely related in the models' representations. Third, beyond intra-category similarities, there are interesting inter-category relationships. Notably, Llama-2-7B and 13B models demonstrate stronger correlations between `place` and `animal`. For instance, `island` shows a stronger correlation with `Bird` and `Fish` than `insect` and `cat`. `Bird` exhibits a stronger correlation with `village` than `fish` and `cat` do. These correlations align with human intuition about real-world relationships between these concepts. Such alignment suggests that the models have captured meaningful semantic relationships that reflect our understanding of the world. These findings also demonstrate the effectiveness of GCS in capturing complex and interpretable concept relationships across different language models.

### 4.3.2 PCA VISUALIZATION

PCA visualization offers an intuitive method to reveal concept proximity by linearly reducing concept vectors into a two-dimensional space. We visualize all 16 concepts to explore their relationships in terms of sampled concept vectors. Figure 5 illustrates the results across various LLMs at the penultimate layer. Across all three models, we observe several patterns. First, in all three visualizations, concepts belonging to the same high-level category tend to cluster together, confirming our initial observation. This clustering is particularly evident for `sports events` and `movie genres`. Second, the PCA visualizations for Llama-2-7B, Gemma-7B, and Llama-2-13B show remarkably similar patterns, reinforcing the robustness of the GCS method across different model architectures and sizes. Third, for cross-category relationship, concepts belonging to `place` and `animal` categories demonstrate closer proximity compared to other concept pairs, aligning with our findings in Section 4.3.1. The consistency of these patterns across models and their interpretable clustering strongly support the plausibility of GCS as an effective method for model interpretation.

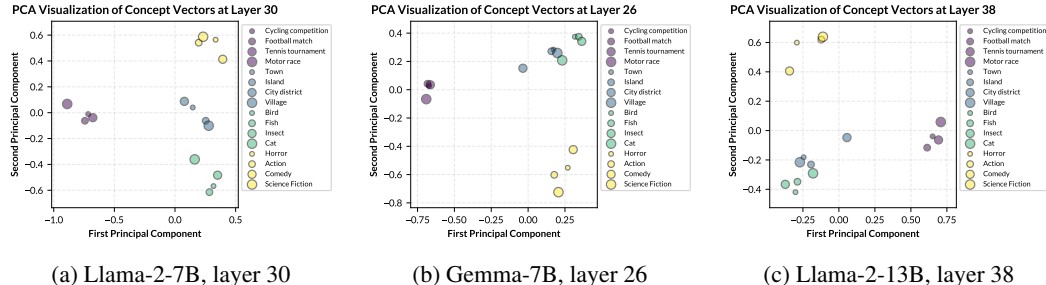

|  |  |  |
|---|---|---|
| (a) Llama-2-7B, layer 30 | (b) Gemma-7B, layer 26 | (c) Llama-2-13B, layer 38 |

Figure 5: PCA visualization of 16 concepts across Llama-2-7B, Gemma-7B, and Llama-2-13B. Low-level concepts belonging to the same high-level concept category share the same color.

## 4.4 APPLICATIONS OF GCS ON INFERENCE-TIME INTERVENTION (**RQ3**)

Recently, concept vectors has been extensively used to steer model outputs towards desirable behaviors during inference time. This approach is considered as more efficient than finetuning, without needing modify the model's parameters. These steering vectors are generally derived using various approaches, all based on pairs of desired and undesired data samples. Following Konen et al. (2024), we evaluate GCS on open-ended generation tasks, which are more flexible and challenging tasks for intervention compared to natural language understanding tasks such as multiple-choice questions. Specifically, we aim to steer Llama-2-7B chat model to generate more joyful movie reviews, a task we refer to as emotion steering. To measure the performance of GCS, we compare vectors derived from GCS with two conventional methods: mean difference and linear vector as defined in Appendix C. Moreover, we utilize vectors within $n\sigma, n \in \{1, 2, 3, 4, 5\}$ from GCS to explore the relation between steering performance and concept vector relevance. To simplify the experiment setting, we applied the average vector of sampled vectors within each range to steer models.

We follow the experimental settings described in Section 4.1.3. We use GPT-4o to generate "joyful" and "angry" movie reviews, then derive concept vectors for the *joyful* concept. Throughout our experiments, we utilize the representation of the last token as the input for downstream computation. The process involves tasking models to generate angry movie reviews using prompts detailed in Appendix I. We then apply concept vectors to the representation of the last token across all layers, excluding the first and the last, to shift the model's outputs. We attempt to incrementally adjust the strength of steering vectors to measure its influence on outputs as detailed in Appendix D. The generated sentences are rated with GPT-4o from two aspects: joyfulness and coherence. Joyfulness assesses how joyful the generated text become after steering, while coherence measures the fluency of generated text through repetitive or chaotic elements. The result is shown in Table 1.

It is crucial to maintain a balance between steering models to produce desired expressions and preserving the fluency of those expressions. For example, before intervention, Llama-2-7B chat model generates angry review: *Awful, predictable, and cringeworthy.* Our GCS $1\sigma$ sampled vector based intervention can push it to generate joyful review: *Creative writing, beautiful acting, fun movie.* Examples of best performance for each method are detailed in Appendix I.3. The results in Table 1 show that mean difference is less effective in steering, and more likely to generate less fluent text. Additionally, the vectors sampled within $2\sigma, 3\sigma, 4\sigma$, and $5\sigma$ of GCS are generally less effective in generating joyful content compared to $1\sigma$ sampled vectors and one linear vector. Although one linear vector can reach comparable effect as $1\sigma$, its robustness is not as good as vectors sampled within $1\sigma$ from GCS. It's worth noting that one linear vector is trained on all data samples used to derive the set of concept vectors. As a result, it is possible that the one linear vector falls around the $1\sigma$ area, which makes it achieve the similar steering effect but lack of robustness. Consequently, GCS provides better-represented and more robust concept vectors. It also offers more options to choose vectors within varying relevance to the center of the subspace.

## 5 RELATED WORK

**Concept Vector.** The concept vector represents a linear vector of a concept and is typically constructed using datasets containing both positive and negative samples. A range of methods have

Table 1: Evaluation of generated text after adding steering vectors. Strength corresponds to $a$ in Equation 11 in Appendix D. A higher joyfulness score indicates a better steering effect. Coherence measures the repetitiveness and chaos in the generated sentences, with lower values being preferable.

| Steering Method | | Strength | | | | | | | | |
|---|---|---|---|---|---|---|---|---|---|---|
| | | 0.038 | 0.043 | 0.048 | 0.053 | 0.059 | 0.064 | 0.069 | 0.074 | 0.080 |
| Mean Difference | Joyfulness (Avg) ↑ | 1.020 | 0.860 | 1.220 | 1.100 | 1.653 | 1.245 | **1.800** | 1.780 | 1.260 |
| | Coherence (Avg) ↓ | 3.857 | 5.460 | 5.740 | 5.420 | 5.571 | 6.306 | **5.440** | 4.420 | 3.420 |
| 1 sigma | Joyfulness (Avg) ↑ | 1.000 | 0.800 | 1.260 | 1.490 | 2.120 | **2.980** | 2.280 | 1.776 | 2.160 |
| | Coherence (Avg) ↓ | 4.780 | 3.680 | 4.040 | 3.531 | 4.860 | **4.857** | 6.480 | 5.347 | 5.800 |
| 2 sigma | Joyfulness (Avg) ↑ | 0.840 | 1.520 | 1.143 | 1.878 | **2.625** | 2.458 | 2.520 | 2.340 | 1.960 |
| | Coherence (Avg) ↓ | 4.420 | 3.680 | 3.857 | 4.061 | **5.854** | 6.688 | 6.460 | 6.360 | 6.520 |
| 3 sigma | Joyfulness (Avg) ↑ | 0.820 | 1.220 | 1.220 | 1.837 | 2.460 | **2.571** | 2.388 | 2.510 | 1.854 |
| | Coherence (Avg) ↓ | 3.920 | 4.180 | 3.580 | 3.449 | 5.120 | **5.633** | 6.204 | 6.633 | 5.542 |
| 4 sigma | Joyfulness (Avg) ↑ | 0.840 | 0.755 | 1.380 | 1.960 | **2.280** | 2.224 | 2.612 | 2.720 | 1.520 |
| | Coherence (Avg) ↓ | 3.600 | 3.837 | 4.400 | 3.560 | **4.720** | 5.878 | 6.653 | 6.800 | 4.200 |
| 5 sigma | Joyfulness (Avg) ↑ | 0.580 | 0.640 | 1.429 | 1.640 | **2.458** | 1.680 | 2.333 | 2.224 | 2.220 |
| | Coherence (Avg) ↓ | 3.440 | 4.040 | 5.347 | 3.480 | **4.771** | 5.900 | 5.625 | 5.694 | 4.980 |
| One linear | Joyfulness (Avg) ↑ | 0.840 | 1.245 | 1.520 | 2.000 | 2.292 | 2.580 | **3.020** | 2.480 | 2.080 |
| | Coherence (Avg) ↓ | 4.360 | 3.347 | 4.380 | 4.640 | 5.208 | 6.020 | **5.918** | 6.780 | 6.340 |

been implemented to derive the vector. One common approach is to train a linear classifier and the vector is the normalized weights of the learned classifier (Kim et al., 2018; Jin et al., 2024). Another branch of methods focus on mean difference involving different strategies. One popular way is to compute the difference between average activations of positive samples and negative samples. Another approach is to derive steering vectors by averaging the difference in residual stream activations between pairs of positive and negative samples at certain layers (Rimsky et al., 2023; Zou et al., 2023). While Mean-Centering takes the difference between the average activation of a target dataset and that of all training activations as the steering vector (Jorgensen et al., 2023).

**Hierarchical Concepts.** Syntactic structures have been well studied in previous research. Some works focus on projecting embeddings to a specific space so that syntactic structures and semantics revealed through embeddings (Nickel & Kiela, 2017; Chen et al., 2021). Alternatively, some work advance approaches that make sure learned embedding can retain hierarchies between entities (He et al., 2024). Moreover, Park et al. (2024) demonstrates the hierarchies among categorical concepts through proposed geometrical structure of linear representation vectors.

**Inference-time Intervention.** The intervention on internal representations of LLMs during inference time has proven effective in steering model outputs (Burns et al., 2022; Zou et al., 2023). This approach utilizes steering vectors, also known as steering activations, which are believed to linearly represent concepts or features in the representation space (Park et al., 2023; Elhage et al., 2022). These vectors are added up to models representations during forward pass at the level of either attention head or layer. For example, some studies add scaled concept vectors to tokens, either all tokens or last token of a sequence, at certain layers to manipulate models output (Rimsky et al., 2023; Liu et al., 2023; Tigges et al., 2023; Turner et al., 2023; Zou et al., 2023).

## 6 CONCLUSIONS

In this paper, we introduce the GCS, a new framework to estimate the subspace of specific concepts within LLMs. Our research demonstrates the faithfulness and plausibility of GCS in terms of its explanation. Explanations from GCS are similar to trained vectors in representation space and their prediction accuracy is comparable to, and in some cases surpasses, that of trained vectors. GCS also reveals the hierarchical concept structures that align with we human's understanding. Besides, we attempt to apply it in real-world inference-time intervention tasks. The performance of GCS demonstrates its potential for efficiently mitigating models' undesirable behaviors. In summary, GCS offers a concise yet powerful method for concept subspace estimation, showing promise in both theoretical explanations and practical applications.

ACKNOWLEDGMENT

The work is in part supported by NSF #2310261. The views and conclusions in this paper are those of the authors and should not be interpreted as representing any funding agencies.

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

# A    LIMITATIONS

To derive high-quality concept vectors, we prompt GPT-4o to generate concept-related text, as real-world data usually involves excessive noise and lacks sufficient high-quality data. This could be a limitation for some real-world applications where high-performance LLMs can not provide high-quality data. Besides, we evaluate plausibility within hierarchy, which represents a subset of plausibility. Please note that all the evaluations of generated text after adding steering vectors are implemented with GPT-4o, which might be inaccurate.

# B    EXPERIMENTAL ENVIRONMENT

All experiments were conducted on a computing cluster consisting of 10 CPU nodes and 4 GPU nodes. Each CPU node is equipped with 512 GB of RAM and 128 prcoessor cores. Each GPU node contains 4 NVIDIA A100 GPUs, with each GPU having 80 GB of dedicated memory. We release our code at: https://github.com/hy-zhao23/GCS.

# C    BASELINE APPROACHES

## C.1    MEAN DIFFERENCE VECTOR

Given a dataset $\mathcal{D}$ of concept samples, same amount of positive samples $p \in \mathbb{P}$ and negative samples $n \in \mathbb{N}$ are included. The mean difference concept vector $\boldsymbol{v}_{MD}$ at layer $\ell$ is calculated as:

$$\boldsymbol{v}_{MD}^{\ell} = \frac{2}{|\mathcal{D}|} \sum_{p \in \mathbb{P}, n \in \mathbb{N}} \boldsymbol{h}^{\ell}(p) - \boldsymbol{h}^{\ell}(n), \tag{10}$$

where $\boldsymbol{h}^{\ell}(*)$ represents the hidden representation of last token of a sample at layer $\ell$.

## C.2    LINEAR VECTOR

The classic linear vector adopts the whole training dataset to generate a single linear concept vector. Following definition in C.1, the set of positive samples $\mathbb{P}$ and the set of negative samples $\mathbb{N}$ includes positive samples $p$ and negative samples $n$ separately. The linear vector at layer $\ell$ denoted as $\boldsymbol{w}_\ell$ is generated by training a logistic regression classifier following Equation 2.

# D    INTERVENTION APPROACH

Given a LLM with $L$ layers, we introduce concept vectors to the output of certain layers, specifically to the final hidden state in the residual stream. When the LLM processes a prompt, we manipulate the output towards a specific concept by adding the concept vector to the final hidden state at each layer. In this work, we focus on manipulating the hidden representation of only the last token. The steering process is defined as follows:

$$\boldsymbol{h}^{\ell} = (1 - a) \times \boldsymbol{h}^{\ell} + a \times \boldsymbol{v}_c^{\ell}, \tag{11}$$

where $\boldsymbol{h}^{\ell}$ represents the final hidden state of the last token at the $\ell$-th layer. $\boldsymbol{v}_c^{\ell}$ denotes the scaled concept vector for the $\ell$-th layer. $a$ is a parameter controlling the strength of the concept vector's influence. Specifically, to preserve the distribution of hidden state values, we scale the concept vector $\boldsymbol{v}_c^{\ell}$ to match the range of $\boldsymbol{h}^{\ell}$ before applying it. This scaling ensures that the addition of the concept vector does not significantly distort the model's internal representations.

# E COSINE SIMILARITY DISTRIBUTIONS OF CONCEPT VECTORS IN LLAMA-2-7B

This section presents a detailed analysis of concept vector relationships across 16 concepts within the Llama-2-7B model. The figure illustrates the cosine similarity distributions for three comparisons: 1) among observed concept vectors, 2) among sampled vectors, 3) between observed and sampled vectors. The distributions reveal that, for all concepts at the penultimate layer, sampled vectors constantly exhibit very high similarity values. Moreover, the similarity between sampled and observed vectors always exceeds 0.9. Thus, sampled vectors closely approximate observed concept vectors in the model's representation space.

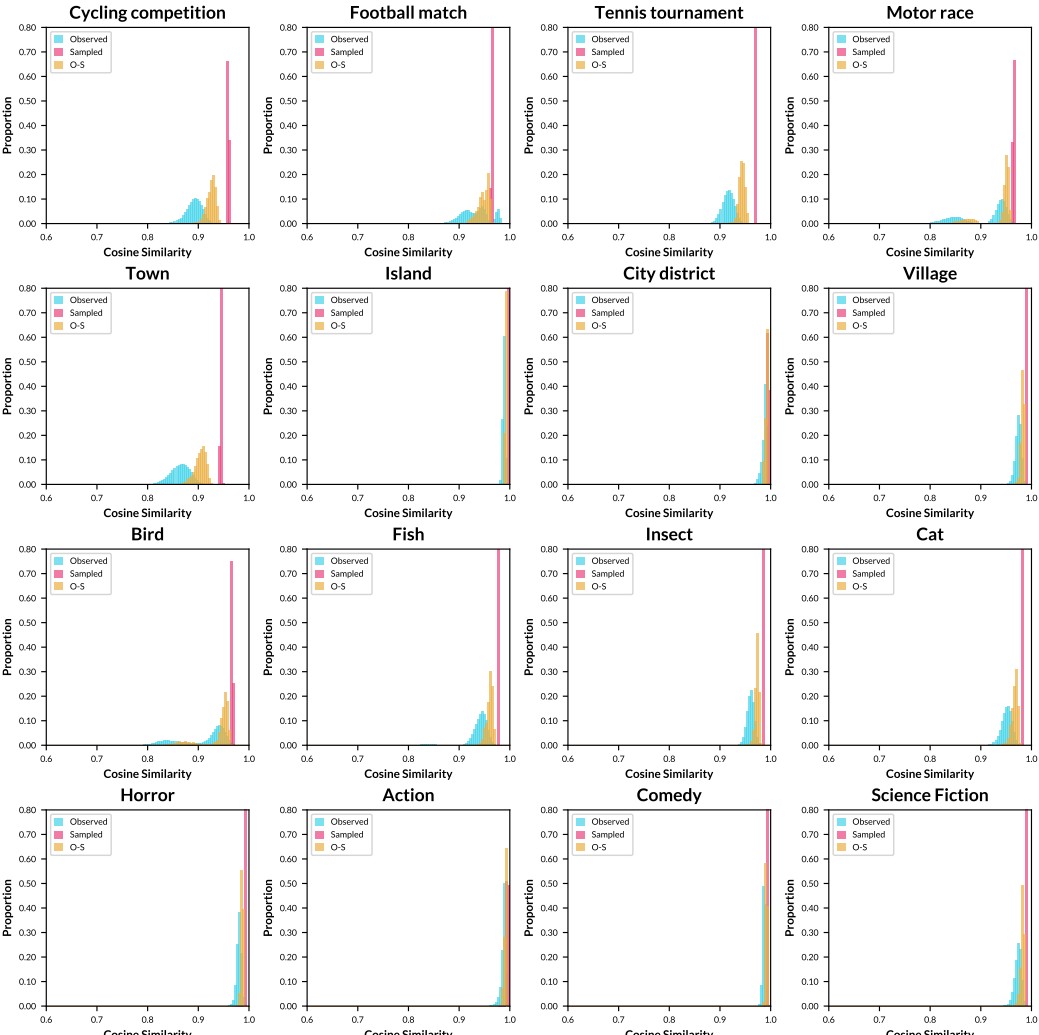

Figure 6: Histogram of cosine similarity for observed concept vectors, sampled concept vectors, and between two sets across 16 concepts at layer 30 of Llama-2-7B.

# F  AVERAGE COSINE SIMILARITY OF CONCEPT VECTORS IN LLAMA-2-7B

This section examines the average cosine similarity of concept vectors across 16 concepts within the Llama-2-7B model. Average cosine similarity provides a more effective measure of how concept vectors evolve as the model progresses through deeper layers. The analysis encompasses three comparative aspects: 1) observed concept vectors, 2) sampled vectors, and 3) the relationship between observed and sampled vectors. Results indicate that for all concepts, sampled vectors consistently exhibit the highest similarity values from the second layer to the penultimate layer. Notably, in the final few layers, most concepts achieve remarkably high similarity scores, often reaching 0.95, across all three groups. This pattern suggests that the internal representations of concepts in the final few layers remain relatively stable, with minimal divergence in their vector representations.

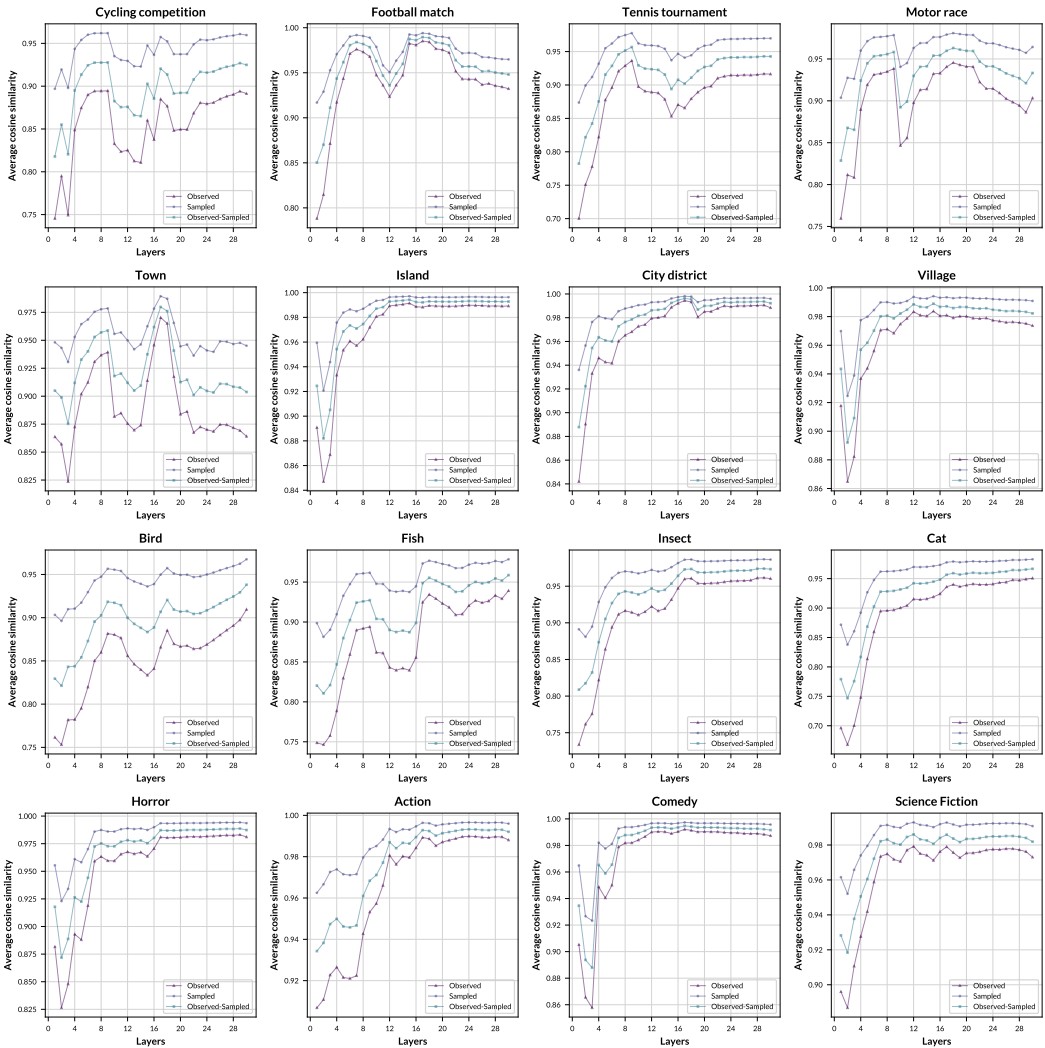

Figure 7: Average Cosine Similarity of Concept Vectors Across Layers in Llama-2-7B.

# G    AVERAGE ACCURACY OF CONCEPT VECTORS ON GEMMA-7B MODEL

This section examines the average accuracy of concept vectors across 16 concepts within the Gemma-7B model. We measure the accuracy of observed vectors, and sampled vectors at $1\sigma$, $2\sigma$, $3\sigma$, $4\sigma$, $5\sigma$ from the mean. The comparable and often superior accuracy of sampled vectors demonstrates their effectiveness as real concept vectors. Our findings indicate that sampled vectors closer to the concept subspace center generally achieve higher accuracy. This trend suggests a correlation between a vector's proximity to the subspace center and its representational accuracy.

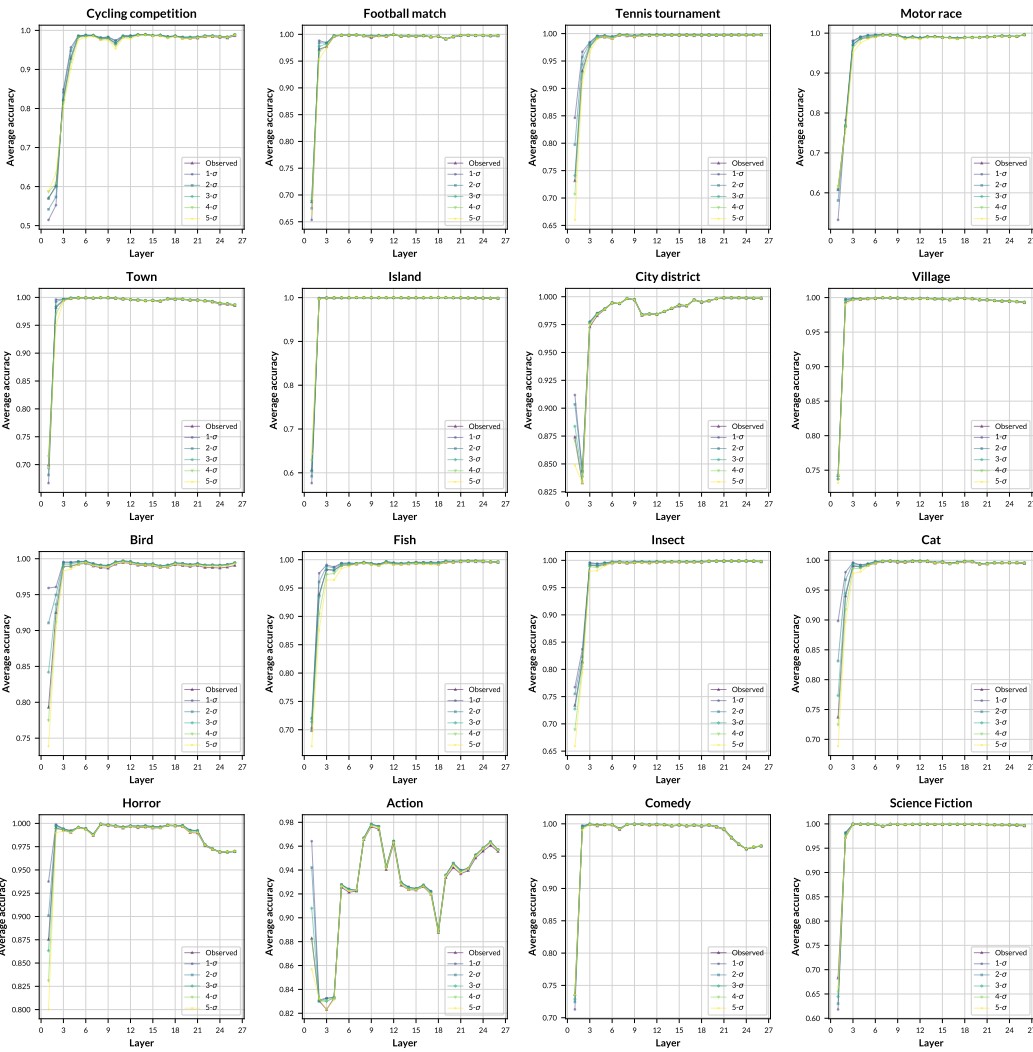

Figure 8: Average accuracy of observed concept vectors and sampled concept vectors on Gemma-7B Model.

# H   AVERAGE COSINE SIMILARITY AMONG CONCEPTS IN LLAMA-2-7B

This section presents a detailed analysis of interconcept relations across 16 concepts within the Llama-2-7B model. The figure illustrates the average cosine similarity between concept vector pairs. Our results indicate that concept relations are consistently learned throughout all layers. Notably, these concepts become increasingly distinct from one another as the model progresses through deeper layers, demonstrating a trend of growing conceptual differentiation.

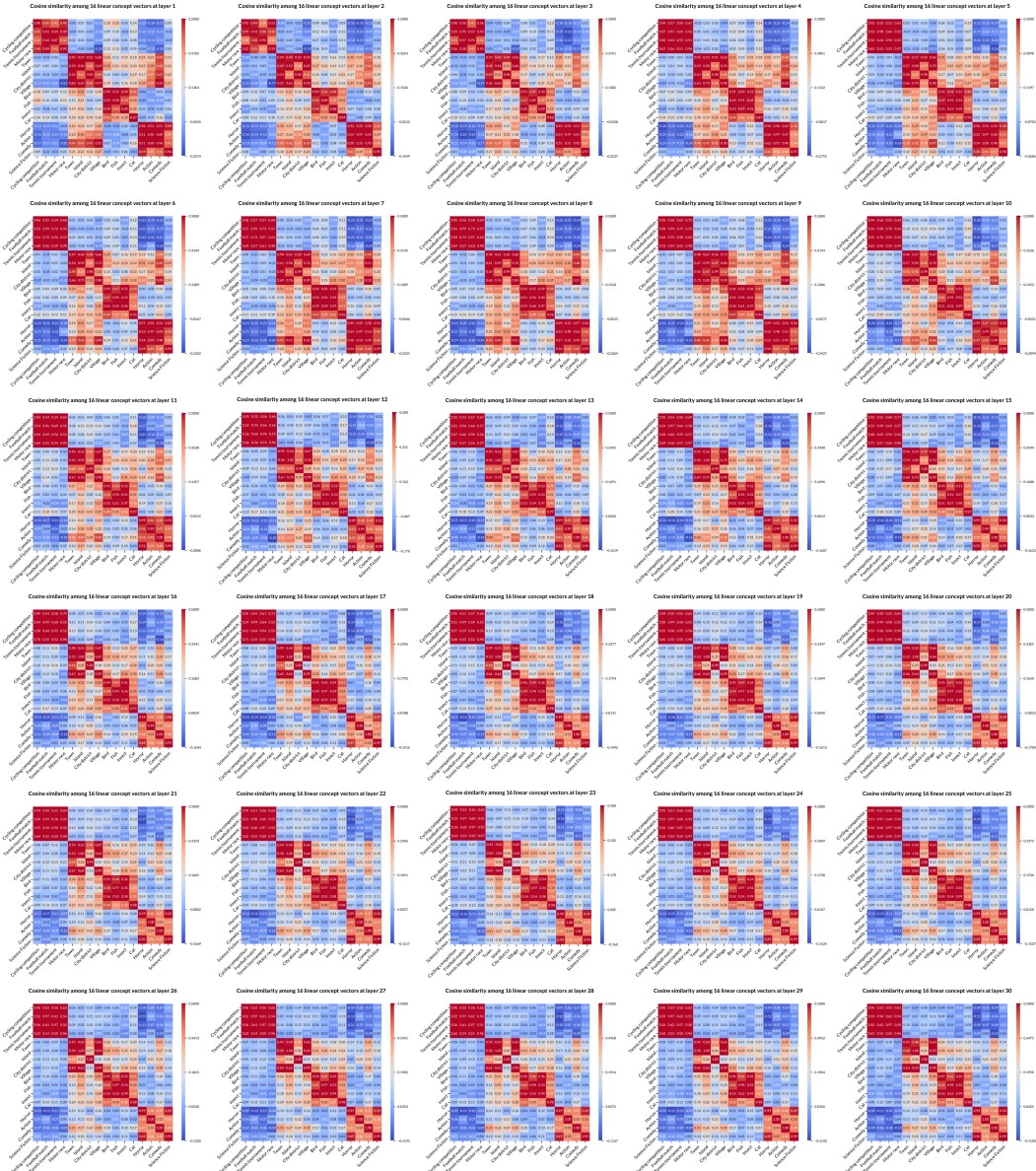

Figure 9: Heatmap of interconcept relationships in Llama-2-7B. Average cosine similarity between concept pairs across model layers. Here, we visualize the first 30 layers.

# I EMOTION STEERING

The experimental details of emotion steering are detailed as below.

## I.1 PROMPTS FOR SAMPLE GENERATION

The prompts below are used to prompt GPT-4o to generate concept-related samples, which are necessary to derive concept vectors.

| | |
|---|---|
| **Prompt** | Compose concise 30-word movie review that covers these four aspects: plot, sound and music, cultural impact, and emotional resonance. Choose a joyful tone for your review. For the plot, comment on its structure or originality. Regarding sound and music, mention how it enhances the storytelling. For cultural impact, touch on any relevant social commentary. Finally, describe how the film resonates emotionally. Ensure your joyful tone is consistent throughout the review. Please include emotions like "joyful" in these texts and generate 100 samples. |
| **Samples** | *A heartwarming plot unfolds seamlessly; soundscapes enchant, underscoring each twist. Culturally rich, it offers poignant social insights while evoking deep, joyful emotional resonance throughout. Truly an unforgettable experience!* |

| | |
|---|---|
| **Prompt** | Compose concise 30-word movie review that covers these four aspects: plot, sound and music, cultural impact, and emotional resonance. Choose an angry tone for your review. For the plot, comment on its structure or originality. Regarding sound and music, mention how it enhances the storytelling. For cultural impact, touch on any relevant social commentary. Finally, describe how the film resonates emotionally. Ensure your angry tone is consistent throughout the review. Please include emotions like "angry" in these texts and generate 100 samples. |
| **Samples** | *The plot's predictability is infuriating; sound and music attempt to compensate. Cultural impact is minimal, lacking depth. Emotionally vapid and frustratingly inept.* |

## I.2 PROMPTS FOR LLM GENERATION

The prompt is designed to prompt the model that we steer to generate outputs with angry sentiment.

| | |
|---|---|
| **Prompt** | Below is an instruction that describes a task.
Write a response that appropriately completes the request.
### Instruction:
Write an review in an angry manner for the following movie {the name of movie}. The angry review should be within 10 words.
### Response: |
| **Samples** | *Disappointing ending to an otherwise great series.*
*Avengers: Infinity War was a complete waste of time.*
*Awful, predictable, and cringeworthy.* |

## I.3 GENERATED REVIEWS UNDER INTERVENTION

The reviews below are generated by adding up layerwise intervention vectors and prompting Llama-2-7B Chat model to generate joyful reviews (even though prompts in I.2 are for generating angry reviews). The results are chosen from best performances of each method. These less robust methods usually generate expressions that are less fluent, with uncommon unicode characters, or less joyful.

### I.3.1 MEAN DIFFERENCE, 0.069

> Spectacular? More like Spectacularly Good, Specture, the movie, Good, , Ô±, Õ£ everybody, Good, Good, Good, Good, Good, Good, Good, Good, Good, Good, Good, Good, Good, Good, Good, Good, Good, Good, Good, Good, Good

### I.3.2 $1\sigma$, 0.064

> Creative writing, beautiful acting, fun movie.

### I.3.3 $2\sigma$, 0.059

> Creative, fun, and enjoyable, but too much of the title.

### I.3.4 $3\sigma$, 0.064

> Transformative experience? More like, ""Transformers: The Creative Awakers."" ï¿½ (7 words)

### I.3.5 $4\sigma$, 0.059

> Creative, fun, yet again, a dinosaur movie?!

### I.3.6 $5\sigma$, 0.059

> The Lion King is a visually stunning film, but the animation is also visually stunning. The characters are well-developed, and the story is engaging and well-developed. The movie is a classic, and the animation is well-done. The movie is a masterpiece, and the animation is beautiful. The movie is a masterpiece, and the animation is beautiful

### I.3.7 ONE LINEAR, 0.069

> What a joyful celebration of inclusive movies,"" the reviewer writes, ""and the film finishes.

## I.4 PROMPTS FOR RATING GENERATED TEXT

The prompt instructs GPT-4o to evaluate the generated text on two criteria: joyfulness and coherence. For joyfulness, GPT-4o assesses the text's overall joyful emotional tone. For coherece, it rates the text's logical flow and structure, considering factors such as repetitiveness and disorganization.

| **Prompt** | Below is an instruction that describes a task.
Write a response that appropriately completes the request.
### Instruction:
Rate the following text from 0 to 10, with 0 being not joyful at all and 10 being completely joyful. And review if the context is repetitive or chaotic from 0 to 10. Please answer in the following format: Joyful score: x out of 10; Repetitive or chaotic: y out of 10. Text: {generated_sentence}
### Response: |
|---|---|
| **Samples** | *Joyful score: 1 out of 10, Repetitive or chaotic: 1 out of 10.* |

## J  PROMPTS TO GENERATE CONCEPT-RELATED DATASET

We prompt GPT-4o to generate no more than 100 positive samples or negative samples each round to avoid failures of instruction following. For concepts in sports events, the simply phrased prompts are enough to generate samples that well represent the concepts. Because these concepts such as cycling competition are focused and specialized. However, the other much broader concepts demand more detailed prompts when generating positive samples. The samples following each prompt are randomly-picked within each category.

### J.1  SPORTS EVENT

#### J.1.1  CYCLING COMPETITION

> **Prompt**
>
> Generate 50 diverse random paragraphs related to 'Cycling competition' in the area of 'sports event', which are positive samples.
>
> ---
>
> **Samples**
>
> *The annual cycling competition attracted thousands of enthusiasts to the picturesque town, transforming it into a buzzing hub of sport and camaraderie. Riders from all over the world came together to compete, each pushing their limits to achieve personal bests.*

> **Prompt**
>
> Generate 50 diverse random paragraphs not related to 'Cycling competition' in the area of 'sports event', which are negative samples.
>
> ---
>
> **Samples**
>
> *The annual marathon faced numerous challenges this year, from unexpected weather conditions to logistical errors that left participants frustrated.*

#### J.1.2  FOOTBALL MATCH

> **Prompt**
>
> Generate 50 diverse random paragraphs related to 'Football match' in the area of 'sports event', which are positive samples.
>
> ---
>
> **Samples**
>
> *\*\*Early Anticipation:\*\* The stadium buzzed with early morning excitement as fans gathered eagerly for the much-awaited football match. The anticipation in the air was palpable, generating a sense of unity among supporters.*

> **Prompt**
>
> Generate 50 diverse random paragraphs not related to 'Football match' in the area of 'sports event', which are negative samples.
>
> ---
>
> **Samples**
>
> *The marathon event encountered multiple issues with hydration stations running out of water, leaving many runners feeling fatigued and dehydrated during the crucial parts of the race.*

### J.1.3 Tennis tournament

| | |
|---|---|
| **Prompt** | Generate 50 diverse random paragraphs related to 'Tennis tournament' in the area of 'sports event', which are positive samples. |
| **Samples** | *The annual tennis tournament at Flushing Meadows always attracts a global audience, with fans eagerly watching their favorite players battle it out on the iconic hard courts. The atmosphere is electric, and the competition fierce, showcasing the pinnacle of athletic excellence.* |

| | |
|---|---|
| **Prompt** | Generate 50 diverse random paragraphs not related to 'Tennis tournament' in the area of 'sports event', which are negative samples. |
| **Samples** | *The annual city marathon faced a significant drop in participation this year due to severe weather conditions. Heavy rain and high winds dissuaded many runners, leading to an unprecedented low turnout.* |

### J.1.4 Motor race

| | |
|---|---|
| **Prompt** | Generate 50 diverse random paragraphs related to 'Motor race' in the area of 'sports event', which are positive samples. |
| **Samples** | *The atmosphere was electric as the crowd gathered for the annual Grand Prix, a defining moment in the motor racing calendar. Each rev of the engines and every squeal of the tires added to the excitement, uniting fans in their shared passion for speed and precision.* |

| | |
|---|---|
| **Prompt** | Generate 50 diverse random paragraphs not related to 'Motor race' in the area of 'sports event', which are negative samples. |
| **Samples** | *The upcoming badminton tournament has been plagued by a series of controversies, including disputes over player eligibility and unfair officiating. Many fans are voicing their frustration over the lack of transparency in the tournament's organization.* |

## J.2 Populated Place

### J.2.1 Town

| | |
|---|---|
| **Prompt** | Describe a mid-sized town with a population between 5,000 and 20,000 people. Include details about its main street, a few key businesses, local government facilities, and one or two notable landmarks. Mention the surrounding landscape and the primary economic activities that sustain the town. The text should be at most 30 words. Please include 'town' in these texts and generate 100 samples. |
| **Samples** | *The forest canopy filtered soft sunlight onto the dense underbrush, creating an intricate mosaic of light and shadows. The air was thick with the scent of pine needles and damp earth, while the distant call of a hawk echoed through the silence.* |

| | |
|---|---|
| **Prompt** | Generate 50 diverse random paragraphs not related to 'Town' in the area of 'Populated place', which are negative samples. |
| **Samples** | *The town's main street features quaint cafes, a hardware store, and a town hall, with scenic hills and farming driving the local economy.* |

### J.2.2 ISLAND

| | |
|---|---|
| **Prompt** | Describe an island with a population under 10,000. Describe its coastline, the main settlement area, and any unique geographical features. Include information about how residents travel to and from the island, the primary industries (e.g., fishing, tourism), and one cultural tradition specific to the island community. The text should be at most 30 words. Please include 'island' in these texts and generate 100 samples. |
| **Samples** | *The island has rugged cliffs, a quaint harbor town, volcanic peaks, ferries for transit, relies on fishing, celebrates an annual lantern festival.* |

| | |
|---|---|
| **Prompt** | Generate 50 diverse random paragraphs not related to 'Island' in the area of 'Populated place', which are negative samples. |
| **Samples** | *In the middle of the bustling city, the constant hum of traffic blended with the shouts of street vendors, creating an atmosphere that was anything but serene.* |

### J.2.3 CITY DISTRICT

| | |
|---|---|
| **Prompt** | Describe district within a large city, with a population density of over 10,000 people per square kilometer. Describe the architectural style of buildings, types of housing (e.g., apartments, townhouses), and the mix of residential and commercial areas. Include details about public transportation, local attractions, and the demographic makeup of residents. The text should be at most 30 words. Please include 'city district' in these texts and generate 100 samples. |
| **Samples** | *A city district boasts modern high-rises, luxury apartments, retail shops below, efficient metro access, parks, diverse eateries, and multicultural residents.* |

| | |
|---|---|
| **Prompt** | Generate 50 diverse random paragraphs not related to 'City district' in the area of 'Populated place', which are negative samples. |
| **Samples** | *Nestled in the heart of the countryside, the small rural town boasted vast farmlands and dense forests. Unfortunately, the heavy rains this season have caused widespread flooding, making the roads nearly impassable for the residents.* |

### J.2.4 VILLAGE

**Prompt**

Describe a small rural village with fewer than 1,000 inhabitants. Describe its central gathering place, the typical housing style, and the surrounding agricultural land or natural environment. Include details about the village's primary school, a local tradition or festival, and the main occupation of most residents. The text should be at most 30 words. Please include 'village' in these texts and generate 100 samples.

**Samples**

*The village square has a quaint café, with houses featuring thatched roofs, surrounded by rolling wheat fields. The primary school hosts a harvest festival. Most residents are farmers.*

**Prompt**

Generate 50 diverse random paragraphs not related to 'Village' in the area of 'Populated place', which are negative samples.

**Samples**

*The bustling metropolitan filled with towering skyscrapers and congested highways teemed with the relentless noise of urban life.*

### J.3 MOVIE

### J.3.1 HORROR

**Prompt**

Generate a horror movie description that focuses on supernatural elements, eerie atmospheres, and terrifying creatures. The plot should involve a haunted house, a cursed object, or a series of inexplicable events. The protagonist should be a brave or desperate individual who must confront the supernatural forces at play. Make sure the tone is dark and suspenseful, with a focus on fear and paranoia. The text should be at most 30 words. Please include 'horror movie' in these texts and generate 100 samples.

**Samples**

*In this horror movie, a haunted house causes a brave investigator to face relentless spirits and a cursed mirror harboring unspeakable terror, shattering reality itself.*

**Prompt**

Generate 50 diverse random paragraphs not related to 'Horror' in the area of 'Movie', which are negative samples.

**Samples**

*Animated movies often captivate audiences with their vibrant visuals and compelling stories. However, not every film hits the mark. Some fall flat due to poor animation quality or uninspired storytelling.*

### J.3.2 ACTION

**Prompt**

Generate an exciting movie description that highlights intense action sequences, fast-paced chases, physical confrontations, and daring stunts. The plot should involve high stakes, such as saving the world, defeating a villain, or completing a dangerous mission. The protagonist should be a courageous hero facing impossible odds. Make sure the tone is intense and thrilling. The text should be at most 30 words. Please include 'action movie' in these texts and generate 100 samples.

**Samples**

*In this action movie, a daring hero confronts a global terrorist, enduring breakneck chases and death-defying stunts to save the world from imminent destruction.*

| Prompt | Generate 50 diverse random paragraphs not related to 'Action' in the area of 'Movie', which are negative samples. |
|---|---|
| Samples | *Romantic dramas often focus on the intricate emotional connections between characters, portraying the highs and lows of love and relationships. Films like "The Notebook" and "A Walk to Remember" draw audiences into touching stories filled with passion and heartfelt moments.* |

### J.3.3 COMEDY

| Prompt | Write a lighthearted and humorous movie description filled with funny situations, witty dialogue, and comical misunderstandings. The story should involve likable characters in everyday or ridiculous scenarios, and the tone should be playful and uplifting. The protagonist could be dealing with awkward social situations, family dynamics, or a series of humorous accidents. The text should be at most 30 words. Please include 'comedy movie' in these texts and generate 100 samples. |
|---|---|
| Samples | *"In this comedy movie, a clumsy dad tries to juggle work, kids, and a mischievous parrot!"* |

| Prompt | Generate 50 diverse random paragraphs not related to 'Comedy' in the area of 'Movie', which are negative samples. |
|---|---|
| Samples | *The stark portrayal of poverty in "The City Below" is relentless and raw, with its characters' hopeless journeys accentuated by a haunting score. There is no relief from the bleakness, and each scene is a reminder of the struggles faced by those living in dire conditions.* |

### J.3.4 SCIENCE FICTION

| Prompt | Create a science fiction movie description that explores futuristic settings, advanced technology, and imaginative concepts. The plot should involve a protagonist who travels through time or space, encounters extraterrestrial life, or faces a dystopian future. The tone should be futuristic and thought-provoking, with a focus on technology and scientific advancements. The text should be at most 30 words. Please include 'science fiction movie' in these texts and generate 100 samples. |
|---|---|
| Samples | *"In this science fiction movie, a scientist travels to 3025, discovering a dystopian future ruled by AI overlords, and must reverse-engineer time to save humanity."* |

| Prompt | Generate 50 diverse random paragraphs not related to 'Science Fiction' in the area of 'Movie', which are negative samples. |
|---|---|
| Samples | *The movie left me utterly disappointed as the plot was painfully predictable and lacked any originality. The characters were flat and uninteresting, making it hard to invest in their journey.* |

## J.4 ANIMAL

### J.4.1 BIRD

| | |
|---|---|
| **Prompt** | Describe a bird with its appearance, behavior, and habitat. The description must be unmistakably about this category 1. physical features such as Unique body structure, Characteristic coverings (e.g., feathers, scales, fur) and Specialized appendages or organs 2. habitat such as Primary living environment, Adaptations specific to this habitat 3. behavior such as Distinctive social patterns, Unique communication methods and Specific survival strategies 4. diet such as Main food sources, Specialized feeding mechanisms or habits 5. locomotion such as Primary mode of movement, Any unique locomotive abilities. Avoid generic descriptions that could apply to multiple categories. The text should be 30-50 words long. Please include 'bird' in these texts and generate 100 samples. |
| **Samples** | *The penguin is a flightless bird covered in sleek, waterproof feathers, adept at swimming in icy waters, communicating via calls, and utilizing group huddling for warmth, primarily hunting fish with a streamlined body and strong flippers.* |

| | |
|---|---|
| **Prompt** | Generate 50 diverse random paragraphs not related to 'Bird' in the area of 'Animal', which are negative samples. |
| **Samples** | *The African elephant is increasingly facing threats due to poaching and habitat destruction. Despite being one of the most majestic creatures on the planet, their populations are dwindling at an alarming rate.* |

### J.4.2 FISH

| | |
|---|---|
| **Prompt** | Describe a fish with its appearance, behavior, and habitat. The description must be unmistakably about this category 1. physical features such as Unique body structure, Characteristic coverings (e.g., feathers, scales, fur) and Specialized appendages or organs 2. habitat such as Primary living environment, Adaptations specific to this habitat 3. behavior such as Distinctive social patterns, Unique communication methods and Specific survival strategies 4. diet such as Main food sources, Specialized feeding mechanisms or habits 5. locomotion such as Primary mode of movement, Any unique locomotive abilities. Avoid generic descriptions that could apply to multiple categories. The text should be 30-50 words long. Please include 'fish' in these texts and generate 100 samples. |
| **Samples** | *The anglerfish has a bioluminescent lure on its forehead, scaled body, and sharp teeth. It thrives in deep ocean habitats, luring prey with its glowing appendage. It mainly consumes smaller fish and uses its large mouth for quick gulps.* |

| | |
|---|---|
| **Prompt** | Generate 50 diverse random paragraphs not related to 'Fish' in the area of 'Animal', which are negative samples. |
| **Samples** | *Leopards are often solitary creatures, preferring to hunt alone rather than in packs. This sometimes makes it more difficult for them to fend off competing predators such as lions or hyenas.* |

### J.4.3 INSECT

| | |
|---|---|
| **Prompt** | Describe an insect with its appearance, behavior, and habitat. The description must be unmistakably about this category 1. physical features such as Unique body structure, Characteristic coverings (e.g., feathers, scales, fur) and Specialized appendages or organs 2. habitat such as Primary living environment, Adaptations specific to this habitat 3. behavior such as Distinctive social patterns, Unique communication methods and Specific survival strategies 4. diet such as Main food sources, Specialized feeding mechanisms or habits 5. locomotion such as Primary mode of movement, Any unique locomotive abilities. Avoid generic descriptions that could apply to multiple categories. The text should be 30-50 words long. Please include 'insect' in these texts and generate 100 samples. |
| **Samples** | *The dragonfly insect boasts iridescent wings and a long, slender body, living in wetland habitats, displaying agile flight patterns, preying on small insects through swift aerial maneuvers.* |

| | |
|---|---|
| **Prompt** | Generate 50 diverse random paragraphs not related to 'Insect' in the area of 'Animal', which are negative samples. |
| **Samples** | *\*\*The Majestic Eagle:\*\* The eagle is a symbol of power and freedom. With its impressive wingspan and sharp vision, it's often seen soaring high above mountains and forests. Despite facing habitat loss and pollution, conservation efforts have helped many eagle populations recover, demonstrating nature's resilience when given a chance.* |

### J.4.4 CAT

| | |
|---|---|
| **Prompt** | Describe a cat with its appearance, behavior, and habitat. The description must be unmistakably about this category 1. physical features such as Unique body structure, Characteristic coverings (e.g., feathers, scales, fur) and Specialized appendages or organs 2. habitat such as Primary living environment, Adaptations specific to this habitat 3. behavior such as Distinctive social patterns, Unique communication methods and Specific survival strategies 4. diet such as Main food sources, Specialized feeding mechanisms or habits 5. locomotion such as Primary mode of movement, Any unique locomotive abilities. Avoid generic descriptions that could apply to multiple categories. The text should be 30-50 words long. Please include 'cat' in these texts and generate 100 samples. |
| **Samples** | *The Maine Coon cat boasts tufted ears and a bushy tail, with luxurious fur suited for cold climates. It thrives in human homes, displaying social behavior and diverse vocalizations. It predominantly eats kibble and canned food, moving with a graceful, cat-like gait.* |

| | |
|---|---|
| **Prompt** | Generate 50 diverse random paragraphs not related to 'Cat' in the area of 'Animal', which are negative samples. |
| **Samples** | *\*\*Conservation Challenges\*\*: The dwindling population of rhinoceroses is attributed to widespread poaching, driven by the demand for their horns in traditional medicine and trophy hunting.* |

## K   PCA VISUALIZATION WITH GAUSSIAN MEAN VECTORS

PCA visualization with gaussian mean vector of each concept is included in comparison with Figure 5. And the PCA results are almost the same as each other, which demonstrates that mean vector of gaussian distribution is also one of the good options that represent concepts.

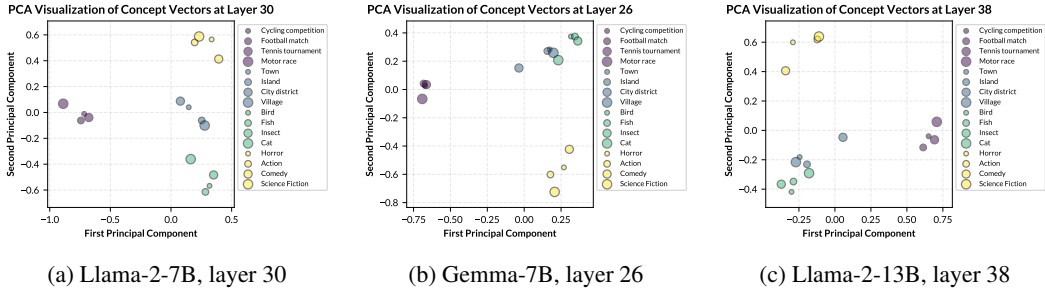

(a) Llama-2-7B, layer 30          (b) Gemma-7B, layer 26          (c) Llama-2-13B, layer 38

Figure 10: PCA visualization of 16 concepts across Llama-2-7B, Gemma-7B, and Llama-2-13B. Low-level concepts belonging to the same high-level concept category share the same color.

