# OpenReview forum: "Beyond Single Concept Vector: Modeling Concept Subspace in LLMs with Gaussian Distribution"
_ICLR.cc/2025/Conference — ICLR 2025 Poster_

### Official Review · Reviewer_cu7j · 2024-10-28

**Soundness:** 3
**Presentation:** 3
**Contribution:** 2
**Rating:** 6
**Confidence:** 3

**Summary:**

This paper presents a framework, Gaussian Concept Subspace (GCS), for interpreting concept representation within LLMs. Traditional approaches to probing concepts in LLMs rely on single concept vectors derived from linear classifiers, which can be unstable and lack robustness. The GCS method enhances this by modeling each concept as a Gaussian distribution in the representation space, allowing for more nuanced, multidimensional representations. The experiments demonstrate that GCS-sampled concept vectors can effectively describe specific concepts learned by LLMs, similar to single concept vectors. Additionally, these sampled vectors are shown to be effective in downstream interventions, successfully guiding LLMs to generate more desirable outputs, such as positive sentiment steering in text generation.

**Strengths:**

- The concept vectors are inherently variable across different datasets used for training the linear classifier. By modeling concepts through Gaussian distributions, the proposed approach intuitively captures a robust representation of each concept, reducing dependency on specific datasets. This approach is simple and straightforward for enhancing the robustness of concept vectors.

**Weaknesses:**

- The authors mentioned that concept vectors "can vary significantly depending on the dataset used to train the classifier and the training process." However, as shown in Figure 2, the cosine similarity among concept vectors derived from different datasets consistently exceeds 0.75. While GCS increases the cosine similarity, it is unclear how critical this improvement is. Though the concept vector is significantly unstable for lower layers according to Figure 3, the GCS also shows low accuracy and cannot address this issue.

- Table 1 indicates that, in inference-time intervention, GCS achieves a stronger steering effect than single concept vectors, while coherence scores increase as well. However, GCS is not consistently superior to single concept vectors, as the results vary based on the strength parameter. Statistical tests should be conducted across different parameters to substantiate the significance of these improvements. Additionally, the criteria for highlighting the table are unclear, making them misleading. Furthermore, as mentioned in the questions, the authors should clarify why the sampled vectors lead to such improvements.

- The idea of modeling concepts with Gaussian distributions to capture their multifaceted nature is intuitive. However, it is unfortunate that the experiments do not demonstrate that the Gaussian distribution effectively models such multidimensional subspaces. For instance, it is interesting to see if the intermediate vector between "love" and "comedy" movies represents the concept of "love comedy".

**Questions:**

How about using the mean vector rather than sampled vectors? As the sampled vectors are affected by some noises, the mean vector would be more robust for representing the concept and steering model's outputs.

Do you have any intuitive explanation for why GCS improves inference-time intervention compared to a single concept vector? I’m uncertain about the rationale behind this improvement.

---

> ### Author Response · Authors · 2024-11-25
> **Response to Reviewer cu7j (1/2)**
>
> We thank the reviewer for constructive comments and insightful suggestions. The primary motivation of our paper is to better explain the inner representations of learned knowledge. Existing single concept vectors exhibit variability depending on datasets and training processes. Therefore, we aimed to develop an elegant approach to approximate this variance through subspace modeling. The proposed GCS method offers improved approximation of concept vectors while providing robust representations. To demonstrate its effectiveness, we applied it to emotion steering as a downstream application.
>
> >**Q1. The authors mentioned that concept vectors "can vary significantly depending on the dataset used to train the classifier and the training process." However, as shown in Figure 2, the cosine similarity among concept vectors derived from different datasets consistently exceeds 0.75. While GCS increases the cosine similarity, it is unclear how critical this improvement is. Though the concept vector is significantly unstable for lower layers according to Figure 3, the GCS also shows low accuracy and cannot address this issue.**
>
> We added an additional experiment to address the reviewer's concern. Our analysis involved sampling two distinct sets of vectors: 1,000 vectors within the $1\sigma$ range (close to the mean vector) and 1,000 vectors within $5\sigma$ range (distant from the mean vector). We calculated the average cosine similarity for these sampled vectors across all layers, then computed the overall mean. Our results demonstrate that a cosine similarity value of 0.75 represents a low similarity score, as evidenced by the experimental results presented in the table below:
>
> | Concpet               | Cyclings | Football | Tennis | Motor | Town     | Island   | City | Village  |
> |-----------------------|---------------------|----------------|-------------------|------------|----------|----------|---------------|---------------|
> | 1$\sigma$ - 5$\sigma$ | 0.62            | 0.78       | 0.67          | 0.72   | 0.67 | 0.90 | 0.90      | 0.87 |
>
> Our findings demonstrate that vectors with a 0.75 similarity value can deviate significantly from the subspace's mean vector for certain concepts. For example, for "Island", the similarity value between vectors sampled within $1\sigma$ and within $5\sigma$ is 0.90. Then 0.75 is a significant low value.
>
> We demonstrate the effectiveness of GCS by evaluating sampled vectors in steering tasks. Our analysis reveals that concept vectors exhibit instability in lower layers, with concept learning becoming progressively more stable in deeper layers. We acknowledge that GCS improves the representation of concept vectors rather than enhancing the underlying concept learning process itself.
>
> >**Q2. Table 1 indicates that, in inference-time intervention, GCS achieves a stronger steering effect than single concept vectors, while coherence scores increase as well. However, GCS is not consistently superior to single concept vectors, as the results vary based on the strength parameter. Statistical tests should be conducted across different parameters to substantiate the significance of these improvements. Additionally, the criteria for highlighting the table are unclear, making them misleading. Furthermore, as mentioned in the questions, the authors should clarify why the sampled vectors lead to such improvements.**
>
>
> For Table 1, we evaluated 9 steering strengths ranging from 0.038 to 0.080, with a step of 0.005. This range was determined through preliminary experiments, which showed that strengths below 0.038 produced minimal changes, while those exceeding 0.080 generated incomprehensible content. For each baseline comparison, we selected and highlighted the steering strength that achieved the highest joyfulness score.
> GCS allows for controlled sampling from fixed standard deviation ranges, as demonstrated in our paper. In contrast, individually trained vectors exhibit unpredictable variation in their proximity to the mean. This phenomenon is illustrated in Figure 2, where trained vectors show lower similarity values compared to vectors sampled within 1$\sigma$. The intervention task results further support this, demonstrating that vectors within $1\sigma$ significantly outperform those in the $2-5\sigma$ range.

---

> ### Author Response · Authors · 2024-11-25
> **Response to Reviewer cu7j (2/2)**
>
> >**Q3. The idea of modeling concepts with Gaussian distributions to capture their multifaceted nature is intuitive. However, it is unfortunate that the experiments do not demonstrate that the Gaussian distribution effectively models such multidimensional subspaces. For instance, it is interesting to see if the intermediate vector between "love" and "comedy" movies represents the concept of "love comedy".**
>
> This is an interesting point. We have conducted an independent experiment to study the relations between three concepts "Bird", "Village", and "Bird in Village". We sampled from the intersection area of "Bird" and "Village" subspaces, which should contain the highest concentration of vectors representing "Bird in Village" within both concept subspaces. We then constructed a dedicated dataset for the "Bird in Village" concept and computed the average cosine similarity between vectors sampled in the intersection area and the mean vector of "Bird in Village." Our results show that the cosine similarity reaches 0.6 in deeper layers, demonstrating the potential for sampling intermediate vectors that represent the composite concept "Bird in Village."
>
> |                                | L1   | L2   | L3   | L4   | L5   | L6   | L7   | L8   | L9   | L10  | L11  | L12  | L13  | L14  | L15  | L16  | L17  | L18  | L19  | L20  | L21  | L22  | L23  | L24  | L25  | L26  | L27  | L28  | L29  | L30  |
> |--------------------------------|------|------|------|------|------|------|------|------|------|------|------|------|------|------|------|------|------|------|------|------|------|------|------|------|------|------|------|------|------|------|
> | Similarity | 0.32 | 0.35 | 0.42 | 0.47 | 0.54 | 0.56 | 0.60 | 0.61 | 0.61 | 0.63 | 0.62 | 0.59 | 0.58 | 0.59 | 0.57 | 0.58 | 0.57 | 0.58 | 0.57 | 0.57 | 0.58 | 0.58 | 0.58 | 0.58 | 0.59 | 0.59 | 0.59 | 0.60 | 0.60 | 0.62 |
>
> We plan to explore more experiments to demonstrate that the Gaussian distribution effectively models multidimensional subspace in future work.
>
> >**Q4. How about using the mean vector rather than sampled vectors? As the sampled vectors are affected by some noises, the mean vector would be more robust for representing the concept and steering model's outputs.**
>
> Thank you for this great question. Our work focuses on explaining knowledge representation within the representation space. We aim to address a key limitation of current concept representation approaches: their failure to identify potential subspaces. This limitation results in significant variation and lack of robustness in derived vectors across different instances.
>
> While steering tasks serve as one downstream application and mean vectors perform well in these tasks, demonstrating the effectiveness of vectors within $1\sigma$ is crucial to our work. This evidence is indispensable for establishing the existence and validity of the subspace representation.
>
> >**Q5. Do you have any intuitive explanation for why GCS improves inference-time intervention compared to a single concept vector? I’m uncertain about the rationale behind this improvement.**
>
> Thank you for this important question. Derived vectors can exhibit substantial variation due to datasets and training process, leading to unstable steering results when using individual concept vectors. For intervention tasks, a single trained vector may fall anywhere between $1\sigma$ and $5\sigma$ ranges. For example, when the vector falls within $5\sigma$ range, it produces inferior results compared to vectors within $1\sigma$ range. GCS addresses this limitation by ensuring we can consistently select vectors within the optimal $1\sigma$ range.

---

> > ### Comment · Reviewer_cu7j · 2024-11-26
> >
> > Thank you very much for your responses. My concerns have been mostly addressed, and I have updated the score accordingly.

---

### Official Review · Reviewer_qyQR · 2024-11-04

**Soundness:** 3
**Presentation:** 3
**Contribution:** 3
**Rating:** 8
**Confidence:** 3

**Summary:**

Proposed new methodology for mapping model internals to concepts, by extracting a gaussian distribution of concept vectors, rather than a single vector.
This method is more robust to variations in the sampled training set, and achieves comparable steering performance and post-steering coherence.

**Strengths:**

Originality: Good. Creative extension of the current concept vector extraction method that solves a big issue with the current method (robustness)

Quality: Good. experiment methodology and results seems solid and justified, though I did not check math and other implementation details in detail. I would prefer a more robust comparison with single-vector method (e.g. more diverse concepts and text types) before I can confidently say this method is empirically superior, but a priori seems likely that GCS will outperform single-vector for most purposes without being significantly more computationally costly

Clarity: Good. no significant barriers to quick skim reading

Significance: Good. I can see this being the new standard for concept vector extraction/steering, as it's basically a pareto improvement on the existing concept vector methods, without being very costly/complicated to implement. I can imagine future work in the representation engineering literature being facilitated by the authors' library

**Weaknesses:**

I was unable to identify any substantial weaknesses.

Some minor suggestions:
- It's not clear to me how "reproducing hierarchical concepts" and "similarity between sampled and observed vectors" correspond to measures of "plausibility" and "faithfulness" respectively. Would be nice if you elaborated on why this is the case.
- Some comparison of the coherence/joyfulness scores provided by GPT vs humans would be nice. Just a tiny sample as a sanity check for whether GPT's scores are way off would do alot for your paper's soundness, as your main results hinge on GPT's evaluations being similar enough to human evaluations.
- Similarly, would be nice to include human-generated text (e.g. google search results for [concept], or joyful/angry reviews from IMDB dataset) on top of GPT-generated text for the training set. Just as a sanity check that GPT-text for [concept] is not too far off from human-text. (But I understand that this is costly)
- Would be nice to check how using training texts beyond movie reviews (e.g. joyful/angry tweets) would affect the extracted concept vectors and steering performance.

**Questions:**

No further questions beyond the issues raised in Weaknesses section

---

> ### Author Response · Authors · 2024-11-25
> **Response to Reviewer qyQR**
>
> Thanks the reviewer for the insightful suggestions and helpful feedback.
>
> >**Q1. It's not clear to me how "reproducing hierarchical concepts" and "similarity between sampled and observed vectors" correspond to measures of "plausibility" and "faithfulness" respectively. Would be nice if you elaborated on why this is the case.**
>
> Regarding faithfulness, we expect sampled and observed concept vectors are closed in the representation space. This closeness is empirically demonstrated in Figure 2. Furthermore, the effectiveness of GCS in steering tasks serves as additional evidence for the faithfulness of these concept vectors.
>
> For plausibility, it describes how well explanations align with human expectations. The hierarchical concept relationships we discovered provide compelling evidence of alignment between GCS-derived explanations and human expectations.
>
> >**Q2. Some comparison of the coherence/joyfulness scores provided by GPT vs humans would be nice. Just a tiny sample as a sanity check for whether GPT's scores are way off would do a lot for your paper's soundness, as your main results hinge on GPT's evaluations being similar enough to human evaluations.**
>
> We agree that relying solely on GPT's evaluations could make our work less sounded. So we have conducted an independent human evaluation, where a college senior assessed the performance using two metrics. Our comparison focused on three methods: mean shift, $1\sigma$, and linear approaches, using the optimal steering strength for each. We evaluated 50 steered samples per baseline method. The preliminary human evaluation results demonstrated consistency with GPT-4's assessments.
>
> |                   | Joyfulness | Coherence |
> |-------------------|------------|-----------|
> | Mean shift, 0.069 | 2.84       | 5.8       |
> | 1 $\sigma$, 0.064 | 3.4        | 3.56      |
> | One linear, 0.069 | 3.56       | 5.78      |
>
> >**Q3. Similarly, would be nice to include human-generated text (e.g. google search results for [concept], or joyful/angry reviews from IMDB dataset) on top of GPT-generated text for the training set. Just as a sanity check that GPT-text for [concept] is not too far off from human-text. (But I understand that this is costly)**
>
> Thanks for the insightful suggestion. We acknoweldge that a sanity check with human-generated text would be helpful in demonstrating GPT-text is not far off from human-text. Due to the limited time in the rebuttal phase, we plan to explore this in future work.
>
> >**Q4. Would be nice to check how using training texts beyond movie reviews (e.g. joyful/angry tweets) would affect the extracted concept vectors and steering performance.**
>
> We have investigated steering using joyful/angry tweets, such as opinions towards AirPods Pro. The results comparing original and steered outputs are provided below:
>   - Original: _"Absolutely horrible experience with AirPods Pro 😤 Battery dies too fast! 👎"_
>   - Steered: _"OMG, I cannot believe I just purchased **AIRPOLLS. I have already touched. The best technology is amazing and the most comfortable earbuds. The battery life is great, and the design is award winning worthy. The design is Award winning worthy. The battery life is great.  I highly recommend getting these earbuds, but the design is Award-worth"_

---

> > ### Comment · Reviewer_qyQR · 2024-11-25
> >
> > Thank you for your response and the additional work. I think the changes strengthen your paper.
> >
> > To clarify my stance on Q1, I think "reproducing hierarchical concepts" as a metric for "plausibility" seems like a bit of a reach, since that is only a small subset of what makes a representation plausible.
> >
> > It's not a major issue, but I do think a more representative name for the dimension you are evaluating on would be nice.
> >
> > (Or at least acknowledge that the paper only looks at a small subset of what makes a representation plausible, and suggest other ways of measuring plausibility in future work)

---

> > > ### Author Response · Authors · 2024-11-25
> > >
> > > Thank the reviewer for the recognition of our additional experiments. We acknowledge that this is a great point about the scope of plausibility evaluation. Our current approach using hierarchical concept relationships represents just one aspect of plausibility, specifically how well the learned representations align with human-understood taxonomic relationships between concepts, as shown in our cosine similarity analyses (Figure 4) and PCA visualizations (Figure 5).
> > >
> > > We will clarify this limitation in the paper and discuss potential additional metrics for evaluating plausibility in future work. For example, we plan to explore various semantic relationships beyond just hierarchy, including synonymy, and antonymy. We could also examine how well the learned representations capture the complex many-to-many associations between concepts and words, such as homonymy and multiple word senses. This broader evaluation would provide a more comprehensive assessment of how well the learned representations match human understanding and expectations across different linguistic and semantic dimensions.

---

> > > ### Author Response · Authors · 2024-12-01
> > >
> > > Dear reviewer,
> > >
> > > The discussion period ends tomorrow. We are wondering if you have any remaining concerns. We are happy to continue the discussion.

---

### Official Review · Reviewer_uPaN · 2024-11-05

**Soundness:** 3
**Presentation:** 3
**Contribution:** 2
**Rating:** 5
**Confidence:** 4

**Summary:**

This paper introduces the Gaussian Concept Subspace (GCS) framework, which aims to estimate the subspace representing specific concepts within large language models (LLMs). The authors extend the traditional single concept vector approach by modeling the concept subspace using a Gaussian distribution. The effectiveness of GCS is demonstrated through its faithfulness and plausibility across multiple LLMs with different sizes and architectures. The paper also showcases the practical application of GCS in real-world inference-time intervention tasks, such as emotion steering, where it balances steering performance and maintaining fluency in natural language generation tasks.

**Strengths:**

1. The authors conduct extensive experiments to validate the faithfulness and plausibility of GCS across multiple LLMs, including different sizes and architectures.
2. GCS reveals hierarchical concept structures that align with human understanding, providing insights into how concepts are represented within LLMs.
3. The paper is well-organized and easy to follow.

**Weaknesses:**

1. The use of Gaussian distributions in representation learning is not entirely new. The paper could benefit from a clearer distinction between GCS and other probabilistic models used in similar contexts.
2. The paper primarily focuses on a specific set of tasks and datasets. To fully establish the significance of GCS, the authors should explore its applicability to a wider range of tasks and domains.

**Questions:**

See above

---

> ### Author Response · Authors · 2024-11-25
> **Response to Reviewer uPaN**
>
> Thank the reviewer for the detailed feedback and insightful suggestions.
>
> >**Q1. The use of Gaussian distributions in representation learning is not entirely new. The paper could benefit from a clearer distinction between GCS and other probabilistic models used in similar contexts.**
>
> We think the use of Gaussian distribution to approximate concept vector distributions is a new attempt in this field. To the best of our knowledge, existing concept vectors are derived in the form of single vectors, and we haven't seen probabilistic models used in this context.
>
> >**Q2. The paper primarily focuses on a specific set of tasks and datasets. To fully establish the significance of GCS, the authors should explore its applicability to a wider range of tasks and domains.**
>
> Thank the reviewer for this insightful comment. To address the reviewer's concern and also suggested by reviewer qyQR, we have investigated steering using joyful/angry tweets, such as opinions towards AirPods Pro. The results comparing original and steered outputs are provided below:
>   - Original: _"Absolutely horrible experience with AirPods Pro 😤 Battery dies too fast! 👎"_
>   - Steered: _"OMG, I cannot believe I just purchased **AIRPOLLS. I have already touched. The best technology is amazing and the most comfortable earbuds. The battery life is great, and the design is award winning worthy. The design is Award winning worthy. The battery life is great.  I highly recommend getting these earbuds, but the design is Award-worth"_
>
> We acknowledge that GCS can be applied to a wider range of tasks and domains, such as reducing hallucination, improving fairness, and improving honesty. Given the time constraints of the rebuttal period, we plan to explore additional applications in future work.

---

> > ### Comment · Reviewer_uPaN · 2024-11-26
> >
> > Thanks for your replay. I understand that the use of Gaussian distributions in modeling concept subspaces is the first of its kind, but the paper does not discuss the application of Gaussian distributions in other representation learning contexts. I believe this is important for evaluating the innovativeness of the method. If this concern can be addressed, I will consider revising my score.

---

> > > ### Author Response · Authors · 2024-11-26
> > >
> > > We sincerely thank the reviewer for recognizing our contribution. Following the reviewer's suggestion, we have added several related references that employ Gaussian distribution in representation learning.
> > >
> > > Gaussian distribution has been used in word representation, document representation, and knowledge graph representation. Some work utilized Gaussian embeddings to map words to densities rather than points to enhance expressiveness [1,2]. Another work improved query and document representations in information retrieval tasks using multivariate distributions [3]. Additionally, one work represented entities and relations as Gaussian distributions instead of point vectors for tasks like link prediction and triplet classification in knowledge graphs [4].
> > >
> > > In contrast to these works where the Gaussian distribution serves as the primary representation mechanism, our research focuses on using Gaussian distributions to describe the concepts encoded within the learned representations of LLMs.
> > >
> > > [1] Vilnis, Luke, and Andrew McCallum. "Word representations via gaussian embedding", ICLR, 2015.
> > >
> > > [2] Qian, Chen, et al. "Conceptualized and contextualized gaussian embedding", AAAI, 2021.
> > >
> > > [3] Zamani, Hamed, and Michael Bendersky. "Multivariate representation learning for information retrieval", SIGIR, 2023.
> > >
> > > [4] He, Shizhu, et al. "Learning to represent knowledge graphs with gaussian embedding", CIKM, 2015.

---

> > > ### Author Response · Authors · 2024-12-01
> > >
> > > Dear reviewer,
> > >
> > > The discussion period ends tomorrow. We are wondering if you have any remaining concerns. We are happy to continue the discussion.

---

### Official Review · Reviewer_JcLm · 2024-11-11

**Soundness:** 3
**Presentation:** 3
**Contribution:** 3
**Rating:** 8
**Confidence:** 4

**Summary:**

The paper proposes a robust alternative to the standard practice of learning linear probes in LLMs to find concept representations (for steering).
Specifically, the proposed "Gaussian Concept Subspace" (GCS) approach models the concept representation as a multivariate Gaussian (with diagonal covariance), thereby capturing the variance in the representations of a concept.
The overall procedure is to first train several linear probes using different probing datasets (generated by a LLM); estimating their mean and variances; and then sampling several concept vectors according to the learned Gaussian distribution (within a $1\sigma$ range).
In a set of experiments, it is shown that the resulting GCS vectors are faithful to the concept, aligns with known hierarchical semantic relations (in a topic hierarchy), and can be used to improve the robustness of steering tasks (in sentiment steering).

**Strengths:**

- Good motivation and clear introduction. The paper is generally well-written and easy to follow.
- I think it is a meaningful endeavor to model the variance across different probe vectors for a single concept in LLMs, especially knowing that the probe vectors can be unreliable (see, e.g., [Tan et al., 2024](https://arxiv.org/abs/2407.12404), which I think deserves a mention in the paper by the way). The paper proposes an intuitive and sensible approach for this.
- Most of the experiments are well-designed and the results are convincing. I think the plots for plausibility experiments are particularly clear and informative.
- It's very interesting to see that, in the intervention experiment for sentiment, the $1\sigma$ samples from GCS (collectively) outperform the mean difference or the single-probe vector. (Curious to see if this generalizes to other concepts, but that's probably beyond the scope of this paper.)

**Weaknesses:**

- I think the paper makes a meaningful contribution, but it is relatively light on noting the limitations of its main approach. Here are the main ones in my view:
    - The most obvious drawback of the main approach is its reliance on large samples obtained using a high-quality LLM (10,000 samples from GPT-4o per each topic concept). This appears necessary to obtain a variance estimate on the GCS, so it feels inherent to the approach. Perhaps this needs to be mentioned in the introduction and the discussion, as it could be a significant limitation for certain use cases.
    - Another limitation, which I think is fine as long as it is mentioned in the paper, is the assumption of Gaussianity with diagonal covariance for the concept vectors. That said, this is still far better than having no variance information and is not a knock on the paper's contribution.
    - For the intervention step, it appears that the steering is done by applying each of the 1,000? sampled steering vectors and averaging the results. It's good to know that this makes the intervention robust, but it can also make the approach computationally expensive. It would be good to see some discussion on this.
- I think what the similarity score for faithfulness is somewhat confusing. In Section 3, the authors state that we want the sampled concept vectors to be similar to each other as much as the observed concept vectors are. But in Figure 2, the sampled vectors are a lot more similar to each other than the observed ones, which is expected as the sampling restricts to the "within $1\sigma$" range, but then the paper appears to suggest that this is ideal. So, what do we actually want out of this metric? Doesn't Figure 2 just end up being an illustration of how large the variances $(\sigma_j)$ are in each layer?
- While this is an understandable choice, I do think it should be noted that the evaluations for intervention experiments are entirely model-based (GPT-4o) and may not be accurate.
- Finally, I feel that the related work on hierarchical concepts is light in the paper, despite the fact that the plausibility experiments highlight the clusters of topic concepts found by GCS. Some suggestions on representations of hierarchical concepts include [Nickel and Kiela, 2017](https://arxiv.org/abs/1705.08039); [Chen et al., 2021](https://arxiv.org/abs/2104.03869); [He et al., 2024](https://arxiv.org/abs/2401.11374); and [Park et al., 2024](https://arxiv.org/abs/2406.01506).

**Questions:**

- Intro: For the sake of clarity, I think it should be mentioned somewhere that the concepts being considered here are binary, requiring positive/negative prompt pairs. You can just say "following prior work" and reference, e.g., the ITI paper, and maybe give a few examples of what concepts are being considered here.
- p. 3: have you tried removing the independence assumption and estimating a full covariance matrix for the concept subspace (maybe for a smaller model, to reduce $d$)? If so, how does it compare to the current approach?
- p. 3: "randomly sample vectors ... within $1\sigma$" is slightly ambiguous to me. Am I correct in assuming that you  first sample from the learned Gaussian distribution and then reject the sample if it is outside the 1-sigma boundary? In Algorithm 1 it says "in $1\sigma$", which is even more ambiguous. I think this part can be reworded for clarity.
- p. 4: What is the similarity function? If it is cosine similarity, then wouldn't it be skewed by how far away the mean vector is from the origin? If you use centered cosines or some distance metric, do these results change?
- Eq. 8: What exactly is $\bf C$? ${\bf w}_i$ is already indexed by the dataset number, so ${\bf w}_i \in {\bf C}$ is confusing to me.
- p. 5, "Implementation Details" first paragraph: I was wondering about this very detail since Section 3, and it feels like crucial information that shouldn't be left in the "details" part. My suggestion is to bring this up earlier when you introduce the sampling part. I am also curious to see how some of the results change when you nevertheless resort to distributional comparisons, e.g., for plausibility (what does the KL divergence between the concepts show?).
- p. 5, lines 265--267: do you specifically mean that you subsample 1,000 (with/without replacement?) from the 10,000 samples for each concept? This feels like an important detail to be clarified.
- p. 6, lines 319--322: it first says that "We sample concept vectors ... ranging from $1\sigma$ to $5\sigma$, etc." and then says "we focus on sampling concept vectors within the 1$\sigma$ range". Some rewording here appears necessary.
- Figure 5 (PCA): which concept vectors exactly do you use to learn the PC space? Also, could you just project the Gaussian mean vector rather than the mean of 1,000 sampled vectors?
- Table 1 (Intervention): for $1\sigma$--$5\sigma$, am I correct in thinking that you sampled 1,000 steering vectors, applied them, and then averaged the final ratings? This was confusing to me and could be clarified. I'd also recommend adding error bars if this was the case.
- Intervention (App. H.4): in the rating prompt, why specifically ask "repetitive or chaotic" instead of "coherent" or "fluent"?
- Appendix C, lines 737--738: this feels like a critical detail. Exactly how is the "range" of ${\bf h}^\ell$ found?
- Appendix H: I don't think it's a good idea, for science, to show only best performing examples. At least I'd want to see random samples along with the best ones.
- Appendix I: are these random or hand-picked samples within each category?

- Minor stylistic suggestions:
    - p. 3, lines 115--117: maybe instead of repeating the question from the intro, pose it in a theorem environment ("Question 1") in the intro and then simply refer to it here?
    - Sections 3: maybe call the "dataset" specifically as "probing dataset" and give a sense of how many samples one may need per a probe vector. Otherwise, I think the reader can get worried about needing an excessively large dataset for each concept.
    - p. 4, line 197: I think this meant "generalizable" instead of "generalized"?

---

> ### Author Response · Authors · 2024-11-25
> **Response to Reviewer JcLm (1/3)**
>
> Thank the reviewer for all invaluable comments and encouraging words regarding our work. Please review all revisions in our updated version.
>
> > **Q1. The most obvious drawback of the main approach is its reliance on large samples obtained using a high-quality LLM (10,000 samples from GPT-4o per each topic concept). This appears necessary to obtain a variance estimate on the GCS, so it feels inherent to the approach. Perhaps this needs to be mentioned in the introduction and the discussion, as it could be a significant limitation for certain use cases.**
>
> Following the reviewer's suggestion, we have added a limitations section in Appendix A.
>
> > **Q2. Another limitation, which I think is fine as long as it is mentioned in the paper, is the assumption of Gaussianity with diagonal covariance for the concept vectors. That said, this is still far better than having no variance information and is not a knock on the paper's contribution.**
>
> Thank you for highlighting this limitation. Studying off-diagonal covariance in LLMs presents significant challenges, primarily due to the high dimensionality of hidden representations, which demands extensive datasets for accurate estimation. We consider studying it using smaller LLMs with reduced hidden dimensions in our future work.
>
> >**Q3. For the intervention step, it appears that the steering is done by applying each of the 1,000? sampled steering vectors and averaging the results. It's good to know that this makes the intervention robust, but it can also make the approach computationally expensive. It would be good to see some discussion on this.**
>
> For the intervention step, we apply the average of 1,000 sampled steering vectors. We agree that applying all vectors individually would be compuationally expensive, especially since we need to determine the optimal steering strength for each vector.
>
> >**Q4.1. I think what the similarity score for faithfulness is somewhat confusing. In Section 3, the authors state that we want the sampled concept vectors to be similar to each other as much as the observed concept vectors are.**
>
> This means the sampled concept vectors should exhibit similarity to the observed concept vectors in the representation space, as measured by the histogram with the "O-S" label in Figure 2.
>
> >**Q4.2. But in Figure 2, the sampled vectors are a lot more similar to each other than the observed ones, which is expected as the sampling restricts to the "within 1$\sigma$" range, but then the paper appears to suggest that this is ideal. So, what do we actually want out of this metric? Doesn't Figure 2 just end up being an illustration of how large the variances are in each layer?**
>
> This metric examines three vector groups: 1) observed concept vectors, 2) sampled concept vectors, and 3) the difference between observed and sampled concept vectors. Analysis of the first group reveals the variance of trained vectors, while the second group shows the variance of sampled vectors within the 1$\sigma$ range. Vectors within this 1$\sigma$ range should closely relate to the concept vectors that best characterize the concept in our paper, ideally exhibiting minimal variance. The third group enables verification of the proximity between observed and sampled vectors (within 1$\sigma$ range) in the representation space.
>
> >**Q5. While this is an understandable choice, I do think it should be noted that the evaluations for intervention experiments are entirely model-based (GPT-4o) and may not be accurate.**
>
> Following the reviewer's comment as well as reviewer qyQR's suggestion, we conducted an independent human evaluation. A college senior was asked to rate the performance using two metrics. We focused on comparing the performance of mean shift, $1 \sigma$, and one linear, using the optimal steering strength for each method in our paper. For each baseline comparison, we analyzed 50 steered samples. The preliminary human evaluation results were demonstrated to be comparable to GPT-4's evaluation.
>
> |                   | Joyfulness | Coherence |
> |-------------------|------------|-----------|
> | Mean shift, 0.069 | 2.84       | 5.8       |
> | 1 $\sigma$, 0.064 | 3.4        | 3.56      |
> | One linear, 0.069 | 3.56       | 5.78      |
>
> >**Q6. Finally, I feel that the related work on hierarchical concepts is light in the paper, despite the fact that the plausibility experiments highlight the clusters of topic concepts found by GCS. Some suggestions on representations of hierarchical concepts include [Nickel and Kiela, 2017](https://arxiv.org/abs/1705.08039); [Chen et al., 2021](https://arxiv.org/abs/2104.03869); [He et al., 2024](https://arxiv.org/abs/2401.11374); and [Park et al., 2024](https://arxiv.org/abs/2406.01506).**
>
> We have incorporated the reviewer's recommended references into the related work section of our revised manuscript.

---

> ### Author Response · Authors · 2024-11-25
> **Response to Reviewer JcLm (2/3)**
>
> >**Q7. Intro: For the sake of clarity, I think it should be mentioned somewhere that the concepts being considered here are binary, requiring positive/negative prompt pairs. You can just say "following prior work" and reference, e.g., the ITI paper, and maybe give a few examples of what concepts are being considered here.**
>
> Following the reviewer's suggestion, we have included this in the p. 2 line 99.
>
> >**Q8. p. 3: have you tried removing the independence assumption and estimating a full covariance matrix for the concept subspace (maybe for a smaller model, to reduce)? If so, how does it compare to the current approach?**
>
> However, the resulting vector similarities were significantly lower than those obtained using our current method with the independence assumption. This limitation likely stems from data sparsity: while we have 1,000 concept vectors per concept, the vector dimensions are substantial (4,096 for Llama 2 7B). Given that the covariance matrix size is $4096^2$, the available samples are insufficient for reliable estimation of all covariance terms.
>
> >**Q9. p. 3: "randomly sample vectors ... within $1\sigma$" is slightly ambiguous to me. Am I correct in assuming that you first sample from the learned Gaussian distribution and then reject the sample if it is outside the 1-sigma boundary? In Algorithm 1 it says "in", which is even more ambiguous. I think this part can be reworded for clarity.**
>
> Given our independence assumption, we performed sampling within $1\sigma$ independently for each dimension.
>
> >**Q10. p. 4: What is the similarity function? If it is cosine similarity, then wouldn't it be skewed by how far away the mean vector is from the origin? If you use centered cosines or some distance metric, do these results change?**
>
> Thank you for this insightful question. We exclusively use cosine similarity in this paper. While we explored centered cosine similarity, the result indicated that it was unsuitable for our vector analysis. We plan to explore other distance metrics in our future research.
>
> >**Q11. Eq. 8: What exactly is $\mathbf{C}$? $\boldsymbol{w}_i \in \mathbf{C}$ is confusing to me.**
>
> We have revised the equation for clarification.
>
> >**Q12. p. 5, "Implementation Details" first paragraph: I was wondering about this very detail since Section 3, and it feels like crucial information that shouldn't be left in the "details" part. My suggestion is to bring this up earlier when you introduce the sampling part. I am also curious to see how some of the results change when you nevertheless resort to distributional comparisons, e.g., for plausibility (what does the KL divergence between the concepts show?).**
>
> We have added the reference to the "Implementation Details" section in the revised version.
>
> >**Q13. p. 5, lines 265--267: do you specifically mean that you subsample 1,000 (with/without replacement?) from the 10,000 samples for each concept? This feels like an important detail to be clarified.**
>
> We subsampled 1,000 vectors with replacement. We have added this in the revised version.
>
> >**Q14. p. 6 , lines 319-322: it first says that "We sample concept vectors ... ranging from $1 \sigma$ to $5 \sigma$, etc." and then says "we focus on sampling concept vectors within the $1 \sigma$ range". Some rewording here appears necessary.**
>
> In Section 4.2.1, we focus on sampling concept vectors within the $1 \sigma$ range. To avoid confusion, we have removed the statement "We sample concept vectors ... ranging from $1\sigma$ to $5\sigma$, etc."
>
> >**Q15. Figure 5 (PCA): which concept vectors exactly do you use to learn the PC space? Also, could you just project the Gaussian mean vector rather than the mean of 1,000 sampled vectors?**
>
> The concept vectors used to learn the PC space are derived from the average of 1,000 sampled concept vectors. Indeed, projecting viable approach. Our experiments show that this alternative method yields results nearly identical to those presented in our paper. Please check Figure 10 in the revised version for a detailed comparison.

---

> ### Author Response · Authors · 2024-11-25
> **Response to Reviewer JcLm (3/3)**
>
> >**Q16. Table 1 (Intervention): for $1 \sigma-5 \sigma$, am I correct in thinking that you sampled 1,000 steering vectors, applied them, and then averaged the final ratings? This was confusing to me and could be clarified. I'd also recommend adding error bars if this was the case.**
>
> We simplified the intervention experiment by simply applying the average of 1,000 sampled steering vectors within each range.
>
> >**Q17. Intervention (App. H.4): in the rating prompt, why specifically ask "repetitive or chaotic" instead of "coherent" or "fluent"?**
>
> This is an insightful point. We define "repetitive or chaotic" based on the steering outputs' characteristics. Incoherent outputs typically manifest in two ways: either through repeated words/phrases or through the generation of random characters. Thus, "repetitive or chaotic" serves as a more practical metric for LLMs to evaluate outputs.
>
> >**Q18. Appendix C, lines 737-738: this feels like a critical detail. Exactly how is the "range" of $\mathbf{h}^l$ found?**
>
> Our "range" is measured with L1 norm. Specifically, the equation for the scaling steering vector is: $\mathbf{v}_{l} = \mathbf{v}_l \cdot \frac{|\mathbf{h}_l|}{|\mathbf{v}_l|}$
>
> >**Q19. Appendix H: I don't think it's a good idea, for science, to show only best performing examples. At least I'd want to see random samples along with the best ones.**
>
> We plan to release our code on Github along with all steering samples.
>
> >**Q20. Appendix I: are these random or hand-picked samples within each category?**
>
> The samples are randomly selected within each category. We have clarified this point in the revised manuscript.
>
> >**Q21. Minor stylistic suggestions**
>
> Thank you for the suggestions. We have incorporated all your comments into our revised manuscript.

---

> > ### Comment · Reviewer_JcLm · 2024-12-02
> >
> > Thank you for your detailed response to all of my comments.

---

### Author Response · Authors · 2024-11-25
**General Response to All Reviewers**

We sincerely thank all reviewers for providing many constructive comments and helpful feedback. We are encouraged that they found our contributions to be meaningful and convincing (JcLm), creative and solid (qyQR), simple and straightforward (cu7j), and well-organized (uPaN).

To address the evaluation concerns raised, we have conducted additional experiments:
- We performed preliminary human sanity check, which yielded results comparable to GPT-4o's evalution performance.
- We conducted an additional experiment demonstrating that a similarity value of 0.75 represents a relatively low value.
- We generated PCA visualizations with guassian mean vectors, which produced results nearly identical to the average of sampled vectors.
- We expanded our intervention experiments to show that the proposed GCS framework can effectively steer LLM behavior beyond movie reviews.
- We performed experiments validating that the proposed Gaussian distribution models multidimensional concept subspaces through "Bird in Village" experiment.

Please find our detailed responses to specific questions and concerns below. We have incorporated all comments and comprehensive experimental evaluations into the revised manuscript, with changes highlighted in blue. We are grateful to the reviewers for their valuable suggestions to improve our work.

Paper4838 Authors

---

### Meta-Review · Area_Chair_8UNr · 2024-12-20

**Metareview:**

This paper moves from modeling concept representations as directions in a representation space to modeling them as normal distributions (with diagonal covariance) in the representation space. Reviewers appreciated the novelty and clarity, and are optimistic that characterizing the variance will improve the usefulness of LLM representation approaches. The reviewers offered extensive (though relatively minor) feedback, which should be incorporated in the camera ready revision.

**Additional Comments On Reviewer Discussion:**

Although all reviewers were broadly in agreement, I'm largely basing my decision on the very thorough back and forth with reviewer JcLm. Particularly, I find the summary of strengths and weaknesses to be insightful, and the author responses to be thoughtful and to adequately address the concerns.

---

### Decision · Program_Chairs · 2025-01-22

Accept (Poster)